# The Role of Magma Mixing in Generating Granodioritic Intrusions Related to Cu–W Mineralization: A Case Study from Qiaomaishan Deposit, Eastern China

**Huasheng Qi [1,2,3], Sanming Lu [1,*], Xiaoyong Yang [2,3,*], Jianghong Deng [4], Yuzhang Zhou [1], Lili Zhao [1], Jianshe Li [1] and Insung Lee [5]**

1   Public Geological Survey Management Center of Anhui Province, Hefei 230091, China;
    qhs87@mail.ustc.edu.cn (H.Q.); hfzhouyz@163.com (Y.Z.); zhaoliligirl@163.com (L.Z.);
    Lijianshe555@163.com (J.L.)
2   CAS Key laboratory of Crust-Mantle Materials and Environments,
    University of Science and Technology of China, Hefei 230026, China
3   CAS Center for Excellence in Comparative Planetology, University of Science and Technology of China,
    Hefei 230026, China
4   CAS Key Lab of Marine Geology and Environment, Center of Deep Sea Research, Institute of Oceanology,
    Chinese Academy of Sciences, Qingdao 266071, China; jhdeng0507@163.com
5   School of Earth and Environmental Sciences, Seoul National University, Seoul 08826, Korea;
    insung@snu.ac.kr
*   Correspondence: lusanming5101@163.com (S.L.); xyyang@ustc.edu.cn (X.Y.); Tel.: +86-0551-6465-2201 (X.Y.)

**Abstract:** The newly exploited Qiaomaishan Cu−W deposit, located in the Xuancheng ore district in the MLYRB, is a middle-sized Cu–W skarn-type polymetallic deposit. As Cu–W mineralization is a rare and uncommon type in the Middle-Lower Yangtze River Belt (MLYRB), few studies have been carried out, and the geochemical characteristics and petrogenesis of Qiaomaishan intrusive rocks related to Cu–W mineralization are not well documented. We studied two types of ore-bearing intrusive rocks in the Qiaomaishan region, i.e., pure granodiorite porphyry and granodiorite porphyry with mafic microgranular enclaves (MMEs). Age characterization using zircon LA–ICP–MS showed that they were formed almost simultaneously, around 134.9 to 135.1 Ma. Granodiorite porphyries are high Mg# adakites, characterized by high-K calc-alkaline and metaluminous features that are enriched in LILEs (e.g., Sr and Ba) and LREEs, but depleted in HFSEs (e.g., Nb, Ta, and Ti) and HREEs. Moreover, they have enriched Sr–Nd–Hf isotopic compositions (with whole-rock $(^{87}Sr/^{86}Sr)_i$ ratios (0.706666−0.706714), negative $\varepsilon_{Nd}(t)$ values of −9.1 to −8.6, negative zircon $\varepsilon_{Hf}(t)$ values of −12.2 to −6.7, and two-stage Hf model ages $(T_{DM}2)$ between 1.5 and 2.0 Ga). However, compared to host rocks, the granodiorite porphyry with MMEs shows variable geochemical compositions, e.g., high Mg#, Cr, Ni, and V contents and enriched with LILEs. In addition, they have more depleted $I_{Sr}$, $\varepsilon_{Nd}(t)$, and $\varepsilon_{Hf}(t)$ values (0.706025 to 0.706269, −6.4 to −7.4, and −10.6 to −5.7, respectively), overlapping with regions of Early Cretaceous mafic rocks derived from enriched lithospheric mantle in the MLYRB. Coupled with significant disequilibrium textures and geochemical features of host rocks and MMEs, we propose that those rocks have resulted from mixing the felsic lower crust-derived magma and the mafic magma generated from the enriched mantle. The mixed magmas subsequently rose to shallow crust to form the ore-bearing rocks and facilitate Cu–W mineralization.

**Keywords:** magma mixing; adakites; Sr–Nd–Pb–Hf isotopes; LA–ICP–MS U–Pb dating; Qiaomaishan Cu–W deposit; Xuancheng ore district; Middle–Lower Yangtze River Metallogenic Belt

## 1. Introduction

The Middle-Lower Yangtze River Belt (MLYRB) is one of China's most important polymetallic metallogenic belts, characterized by porphyry-skarn-type copper-gold polymetallic deposits and porphyrite-type iron deposits [1–5]. Xuancheng is the site of a newfound ore cluster in the MLRYB (Figure 1), where several copper, molybdenum, and iron deposits were discovered, including the super-large Chating porphyry Cu–Au deposit, the Qiaomaishan skarn Cu–W deposit, the Magushan skarn Cu–Mo deposit, the Changshan skarn Cu–Pb–Zn deposit, and the Shizishan Cu deposit [6,7]. The Qiaomaishan deposit, located in the northeast of the Xuancheng ore cluster and southeast of the Jiulianshan–Xinhezhuang Thrust Fault (Figure 2a), has been regarded as a typical Cu–S skarn deposit within the MLYRB. Recently, scheelite was identified in sulfide ores in the Qiaomaishan deposit [8], which is unusual in the MLYMB compared to other porphyry and skarn deposits.

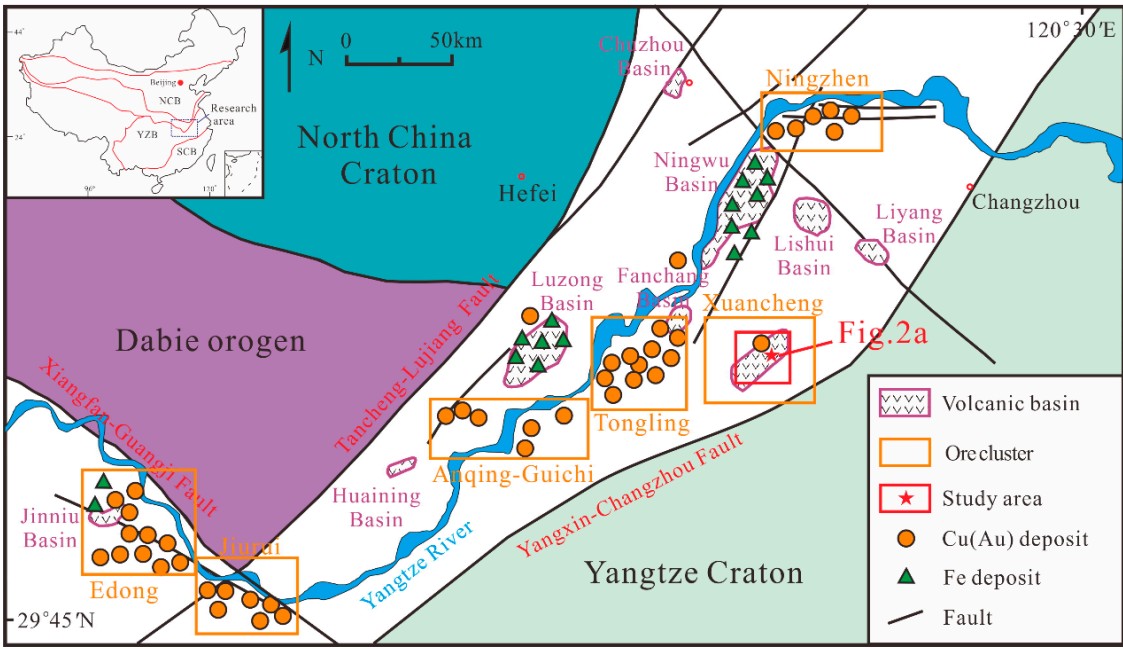

**Figure 1.** Geological scheme, including volcanic basins and ore cluster regions in the Middle-Lower Yangtze River Metallogenic Belt (MLYRB) (modified after [9]).

In contrast with the economically significant W–Cu deposits (e.g., Zhuxi and Dahutang) associated with granitoid in the Jiangnan Tungsten Belt [10–12], Cu–W mineralization in Qiaomaishan is negligible. However, mineralization of Cu–W is rare and uncommon in the MLYRB, making Qiaomaishan a significant example for investigating regional polymetal mineralization. Since Cu–W is an uncommon type of mineralization, few cases have been reported [13], and the geochemical characteristics and petrogenesis of the Qiaomaishan intrusive rocks related to Cu–W mineralization are not well understood. In particular, the Qiaomaishan deposit is not only significant for investigating the regional tungsten mineralization events in the MLYRB but also provides an ideal area to study crust–mantle interaction, due to the abundance of mafic microgranular enclaves (MMEs) in the host's granodiorite porphyry. In this research, we carried out systematic geochemical and comparative studies of two types of ore-bearing intrusive rocks in Qiaomaishan, pure granodiorite porphyry and granodiorite porphyry with MMEs. This included precise zircon LA–ICP–MS U–Pb dating, analysis of the geochemistry of the minerals and rock, and Sr–Nd–Pb–Hf isotopic analyses to reveal the petrogenesis and possible enrichment mechanisms behind Qiaomaishan Cu–W mineralization.

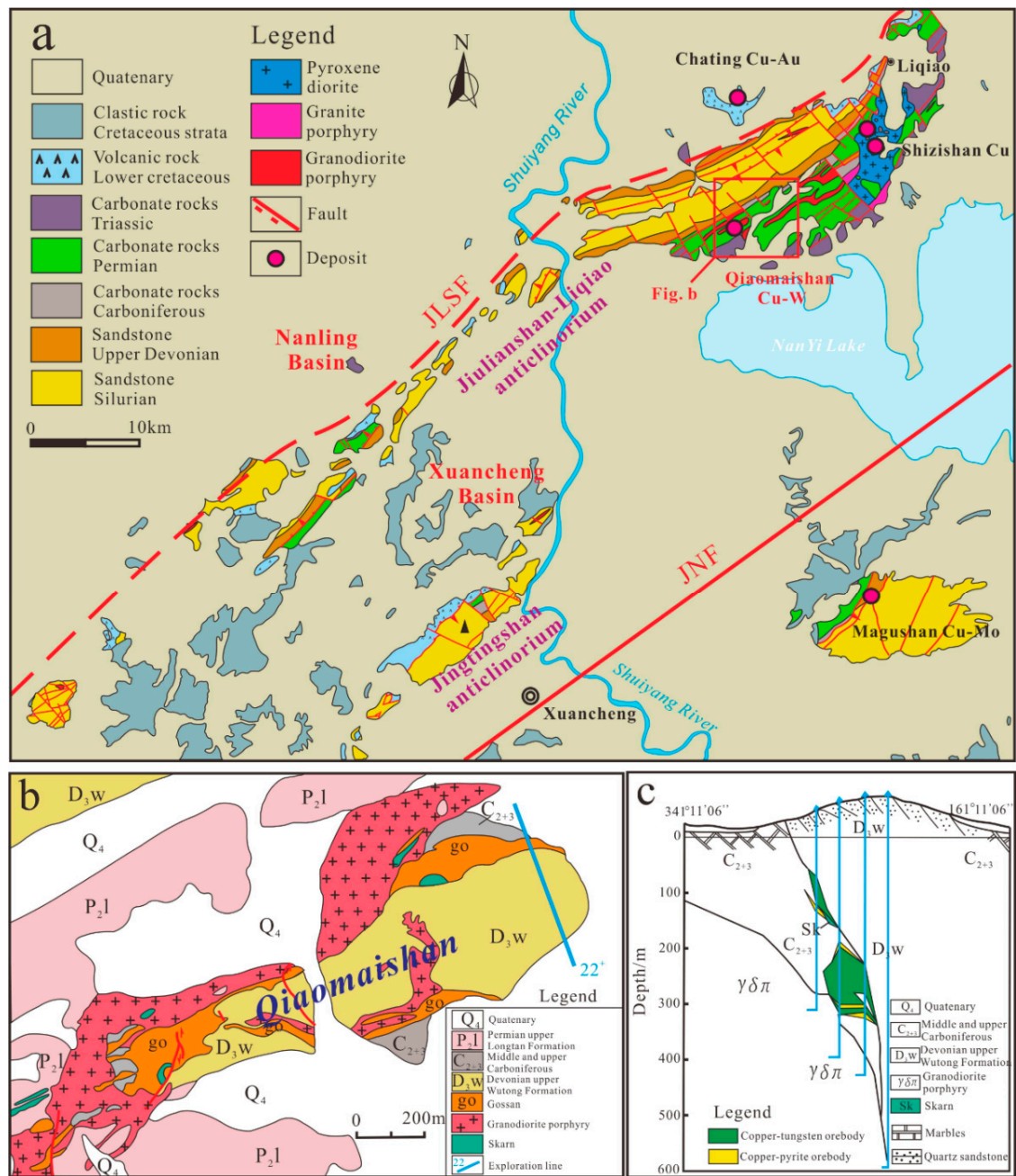

**Figure 2.** (**a**) Geological sketch map of the Xuancheng ore district (modified after [7]); (**b**) Geological map of the Qiaomaishan Cu–W deposit (modified after No.322 Geological Party, Bureau of Geology and Mineral Exploration of Anhui Province); (**c**) Section of the No. 22+ exploration line in the Qiaomaishan Cu–W deposit. Abbreviations: JLSF—Jiulianshan Thrust Fault; JNF—Jiangnan Fault.

## 2. Geological Setting

### 2.1. Regional Geology

The MLYRB is located in the north margin of the Yangtze Craton, adjacent to the southeastern margin of the North China Craton and the Qinling–Dabie Orogenic Belt. It is bounded by the Xiangfan–Guangji Fault (XGF) to the southwest, the Tancheng–Lujiang Fault (TLF) to the northwest and the Yangxin–Changzhou Fault (YCF) in the south and southeast (Figure 1). The metamorphic crystalline basement of the Yangtze Craton consists of ancient amphibolites as well as granulite facies

biotite–hornblende gneisses, Trondhjemite–Tonalite–Granodiorites (TTGs) and a sequence of Mesozoic sedimentary rocks [1]. The exposed stratigraphic sequences consist of shallow marine clastic and carbonate rocks, ranging from the Sinian to the Early Triassic [14], and continental clastic rocks and volcanic rocks from the middle–late Triassic to the Cretaceous [15].

The MLYRB has experienced significant Mesozoic magmatic activity mainly through three major events: (1) ~148–133 Ma, high-K calc-alkaline intrusions (mostly adakite-like; high Sr/Y and La/Yb), widely distributed in uplifted areas [1] and closely connected with the skarn or porphyry Cu–Au deposits [9,16,17] in the MLYRB ore cluster regions (e.g., Tongling, Edong, Anqing, and Jiurui); (2) ~134–129 Ma, sub-alkaline to alkaline volcanic rocks [18–20] formed in faulted volcanic basins, including the Jinniu, Fangchang, Luzong, and Ningwu basins, from SW to NE. (3) A-type granites (~127–124 Ma), distributed along the Yangtze River fault [21–23].

The Xuancheng ore district is located at the eastern Tongling ore cluster district. This district features a new ore cluster distinguished by copper, gold, and molybdenum deposits (Figure 2a) and is mostly covered by Quaternary slope debris and Cretaceous volcanic and clastic red beds of the Zhongfencun– and Xuannan Formation [13,24]. Silurian and Devonian shallow marine sandstones and Triassic carbonate rocks mainly occur at the northeastern part of the Xuancheng region. The Jiulianshan Thrust Fault (JLSF) and the Jiangnan Fault (JNF) are two main NE-trending faults [24]. The distribution of magmatic rocks in this area has been controlled by a series of NW-trending faults and by an NE-trending fold. The Jiulianshan–Liqiao anticlinorium has a thrust nappe structure [25,26] that favors emplacement of a large number of intrusions (e.g., Qiaomaishan, Shizishan, and Changshan). These intrusions are dominantly pyroxene diorite, quartz diorite porphyry, granodiorite porphyry, and granite porphyry, and the volcanic rocks are dacitic pyroclastic rocks and lavas (Figure 2a).

## 2.2. Geology of Deposits

The Qiaomaishan Cu–W deposit, located in the northeast of the Xuancheng ore district (Figure 2a), is a middle-sized skarn-type polymetallic deposit with total estimated resources of 10.7 Mt Cu and 0.83 Mt $WO_3$ [8]. From the oldest to the most recent, the main stratigraphic units in this area include the Upper Devonian Wutong Formation, the Middle–Upper Carboniferous Huanglong and Chuanshan Formation, the Lower Permian Qixia Formation, the Permian Upper Longtan formation, and the Quaternary Holocene covers (Figure 2b). Under the influence of multistage tectonic movements, the Jiulianshan–Liqiao anticline developed a large number of secondary folds and northeast-trending faults accompanied by many secondary faults that control the location and shape of the orebody. The granodiorite porphyry is the main intrusion in this area. This porphyry occurs in the form of stocks or dikes and emplaces into Carboniferous limestone, which is closely related to the skarn Cu–W mineralization in the contact zone (Figure 2b).

Four main Cu–W orebodies were found in the contact zone between Carboniferous limestone and granodiorite porphyry. These orebodies occur mainly in lenticular and stratoid forms (Figure 2c). The main ore minerals are pyrite, chalcopyrite, bornite, magnetite, and scheelite (Figure 3e,f). The chalcopyrite and pyrite show an euhedral or hypidiomorphic granular texture with sizes of 0.2–0.8 mm, and the scheelite normally occurs as fine-grained disseminated crystals with sizes of 0.2–0.4 mm (Figure 3g,h). The alterations of the Qiaomaishan deposit consist of skarnization, carbonation, kaolinization, sericitization, chloritization, and silicification. Liu and Duan [8] classified five main stages of skarn and ore formation according to the mineral assemblages, ore textures/structures, and alteration characteristics of the wall rock: (I) skarn, (II) quartz–magnetite, (III) sulfide, (IV) tungsten mineralization, and (V) carbonate.

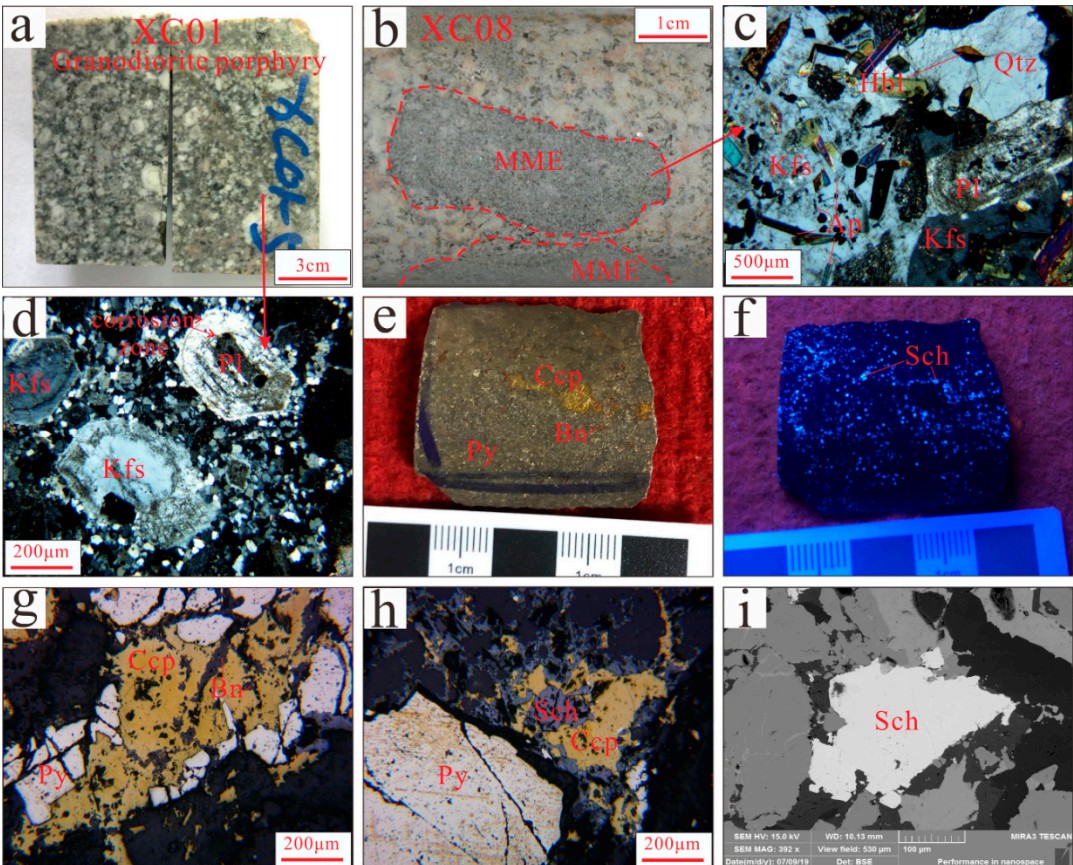

**Figure 3.** Photographs and photomicrographs of the Qiaomaishan Cu–W deposits. (**a**) Granodiorite porphyry; (**b**) granodiorite porphyry with mafic microgranular enclaves (MMEs); photomicrographs of the (**c**) MMEs and (**d**) granodiorite porphyry; (**e**) copper tungsten ores; (**f**) scheelite in the ores under a UV lamp; (**g**,**h**) chalcopyrite, pyrite, and scheelite in the sulfide ores under reflected light; (**i**) representative scheelite under backscattered electron. Bn—bornite; Ccp—chalcopyrite; Py—pyrite; Sch—scheelite; Ap—apatite; Hbl—hornblende; Kfs—K-feldspar; Pl—plagioclase; Qtz—quartz.

## 3. Methods and Sample Descriptions

Based on detailed field investigations, we collected a series of granodiorite porphyry samples from the drill cores, including 7 granodiorite porphyries (ZK22+02, −316 to −450 m) and 5 granodiorite porphyries with MMEs (ZK18+02, −107 to −337 m, Table S1). Particular facies of the granodiorite porphyry host abundant mafic microgranular enclaves (MMEs), mostly 1–3 cm in size (Figure 3b).

We carried out zircon LA–ICP–MS U–Pb dating, EMPA analysis of the plagioclase, and Sr–Nd–Pb–Hf isotopic analyses for two types of intrusive rocks (pure granodiorite porphyry and granodiorite porphyry with MMEs) in Qiaomaishan. The analytical methods are described in detail in Appendix A.

Granodiorite porphyry (GP): Samples are steel gray to dark gray in color with medium- to fine-grained typical porphyritic textures (Figure 3a,d). Phenocrysts form 30–40% of the porphyry's volume. These phenocrysts include K-feldspar, plagioclase, hornblende, and quartz. The groundmass consists of plagioclase, quartz, and melanocratic minerals with a microcrystalline texture (Figure 3d). The K-feldspar and plagioclase phenocrysts are euhedral to subhedral, 0.1–0.4 mm in size, and develop numerous twin crystals that show corrosion zone textures. Notably, some plagioclase phenocrysts display reverse zonal textures (Figure 4a). The hornblende is subeuhedral, with a size of 0.2–0.6 mm; some of the hornblende has suffered chlorite alteration. The quartz phenocrysts are

xenomorphic–granular with a corrosion structure, and some are harbored under corrosion (Figure 4c). The accessory minerals mainly comprise magnetite, apatite, and zircon.

The granodiorite porphyry with MMEs (GPM): The samples consist of host rock and numerous MMEs of a small size (Figure 3b). The petrographic characters of the host rocks are similar to those of the GP, whereas the enclaves are different. The enclaves are dark gray in color, with a fine-grained micropoikilitic texture (Figure 3c), showing faded contact with the host rocks. The mineral compositions are consistent with those of the host granodiorite porphyry, mainly containing plagioclase, K-feldspar, hornblende, quartz, and accessory magnetite, apatite, and zircon. However, the mafic minerals (e.g., hornblende and biotite) occur as small euhedral crystals in K-feldspar and quartz phenocrysts (Figure 4c,d). The hornblende and K-feldspar phenocrysts are euhedral to subhedral with a significantly diablastic texture (Figure 4d,e). In addition, the presence of acicular apatite (Figure 4f) indicates the quenching process of the mafic magma.

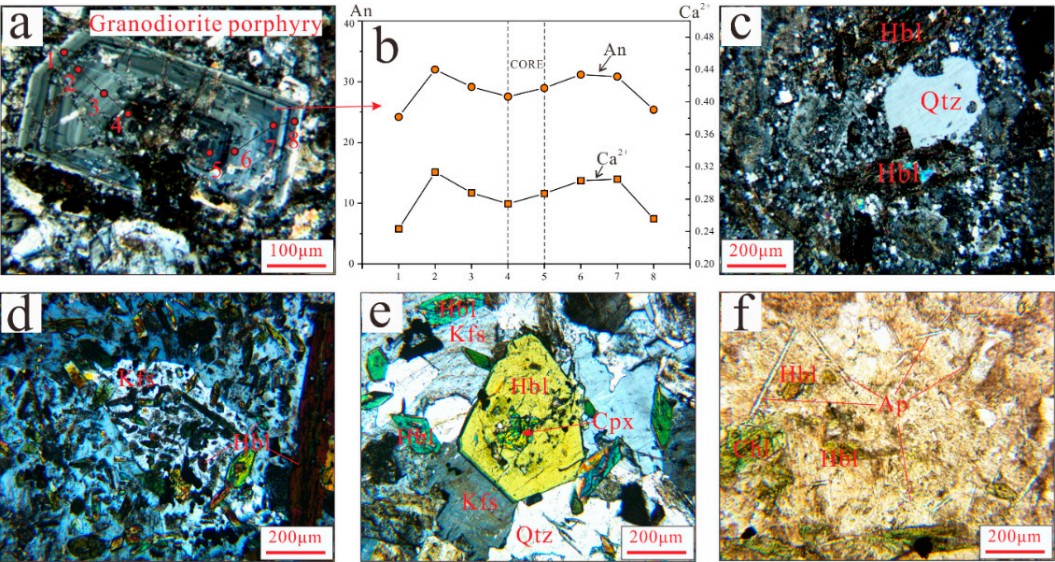

**Figure 4.** Representative disequilibrium textures in the Qiaomaishan intrusions. (**a**) Photomicrograph of the zoned plagioclase showing the spots of electron probe microprobe analyses (MPMA). (**b**) Profiles showing the variation of An (circles) and $Ca^{2+}$ (squares) in the zoned plagioclase. (**c**) Corrosion texture of quartz. (**d**) K-feldspar phenocryst with a microcrystalline texture. (**e**) Hornblende phenocryst with a significantly diablastic texture. (**f**) Minor acicular apatites in mafic enclaves indicating quenching. Ap—apatite; Cpx—clinopyroxene; Hbl—hornblende; Kfs—K-feldspar; Pl—plagioclase; Qtz—quartz.

## 4. Results

Two types of intrusive rocks (GP and GPM) related to Cu–W mineralization in the Qiaomaishan were analyzed. The results for GP and GPM represent the geochemical compositions of the pure granodiorite porphyry and granodiorite porphyry with mafic microgranular enclaves (MMEs). It must be noted that GPM cannot represent MMEs as it is a product made of from a mixture of MMEs and the granodiorite porphyry.

### 4.1. Whole-Rock Geochemistry

The results of the major and trace elements in the Qiaomaishan intrusions are shown in Supplementary Table S1. All values for major-element contents were normalized to 100% on a loss on ignition (LOI) free basis. GP and GPM show large compositional variations and may be classified into quartz monzonite and monzonite–quartz monzonite based on the TAS diagram (Figure 5a), with $SiO_2$ contents ranging from 61.60 to 65.72 wt. % and 58.30 to 63.02 wt. %, respectively. They are all plotted in the high-K calc-alkaline field in the $SiO_2$ vs. $K_2O$ diagram (Figure 5b), with calculated A/CNK values

ranging from 0.79 to 1.01, which indicates metaluminous to peraluminous compositions (Figure 5c). In addition, they have high Mg#, though GPM (2.46–3.48 wt. %, avg. 49) has higher MgO and Mg# than GP (0.9–2.32 wt. %, avg. 45).

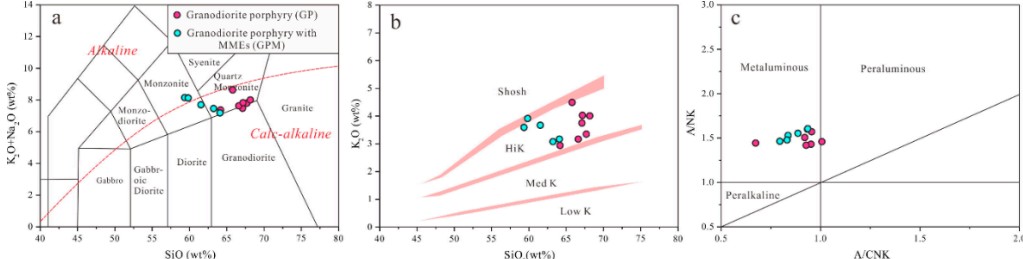

**Figure 5.** Classification diagrams of the lithochemical compositions for the Qiaomaishan intrusions. (**a**) Total alkalis vs. silica (TAS) [27], (**b**) plot of $K_2O$–$SiO_2$ [28], (**c**) A/NK versus A/CNK diagram for the Qiaomaishan granodiorites. The alkaline and sub-alkaline divisions are based on [29].

In the Harker diagrams of major elements (Figure 6), CaO, MgO, $P_2O_5$, $Fe_2O_3^T$, and $TiO_2$ display a negative correlation with an $SiO_2$ increase. However, the lack of a significant correlation between silica contents and $Al_2O_3$, $K_2O$ and $Na_2O$ (Figures 5b and 6) suggests that the fractional crystallization (FC) processes produced only a limited value of silica. The Qiaomaishan intrusions show a negative correlation between $SiO_2$ with Y (Figure 6g), which is inconsistent with the condition that the Y contents in a melt will increase as basaltic to andesitic magma suffers from the FC processes of olivine and pyroxene. Furthermore, there are no correlations between $SiO_2$ with Sr/Y and La/Yb (Figure 6h,i).

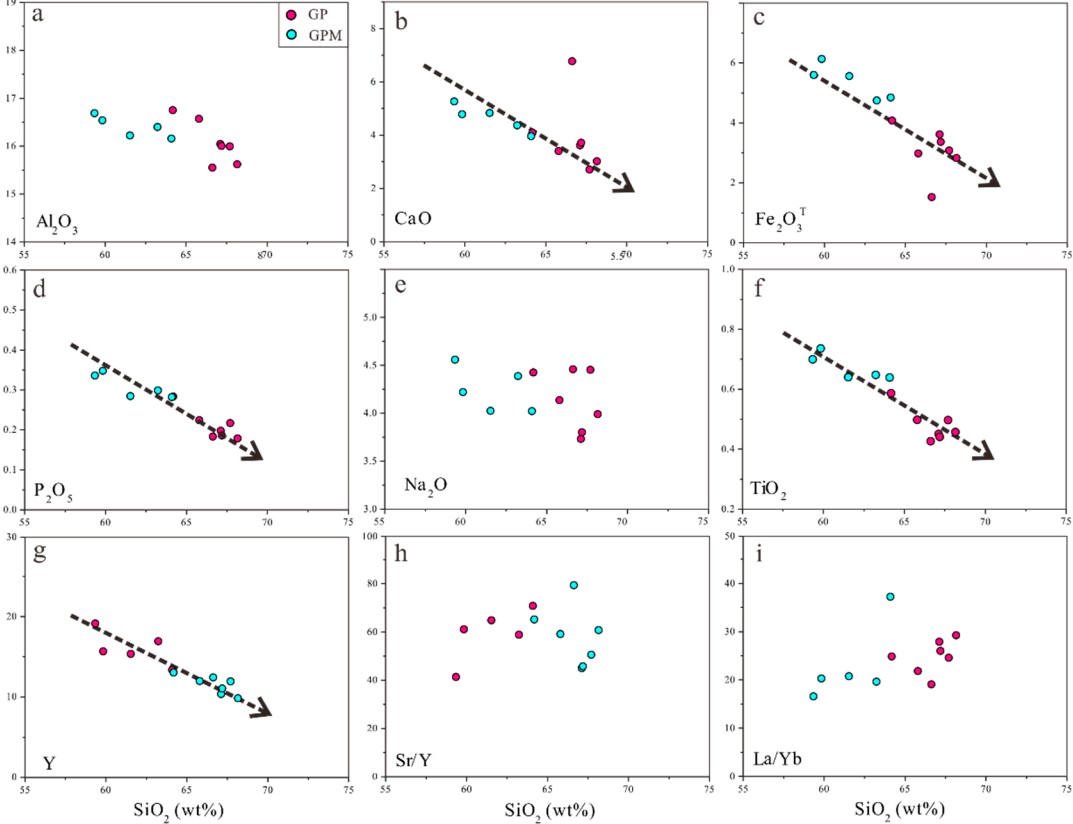

**Figure 6.** Harker diagram of the Qiaomaishan intrusions. Plot of (**a**) $SiO_2$ vs. $Al_2O_3$, (**b**) $SiO_2$ vs. CaO, (**c**) $SiO_2$ vs. $Fe_2O_3^T$, (**d**) $SiO_2$ vs. $P_2O_5$, (**e**) $SiO_2$ vs. $Na_2O$, (**f**) $SiO_2$ vs. $TiO_2$ (**g**) $SiO_2$ vs. Y, (**h**) $SiO_2$ vs. Sr/Y, (**i**) $SiO_2$ vs. La/Yb for the Qiaomaishan intrusions.

Notably, they display high Sr/Y and (La/Yb)$_N$ ratios, with different Y and Yb contents. On the Sr/Y–Y and (La/Yb)$_N$–Yb$_N$ diagrams (Figure 7a,b), the granodiorite porphyry samples (GP) are projected in the area of the adakites. By comparison, samples of the granodiorite porphyry with MMEs are not adakites in Figure 7a, b due to their high Y and Yb contents. Furthermore, Mg# has a positive correlation with Sr contents (Figure 7f), implying that the amount of Sr is determined by the involvement of mantle materials [30].

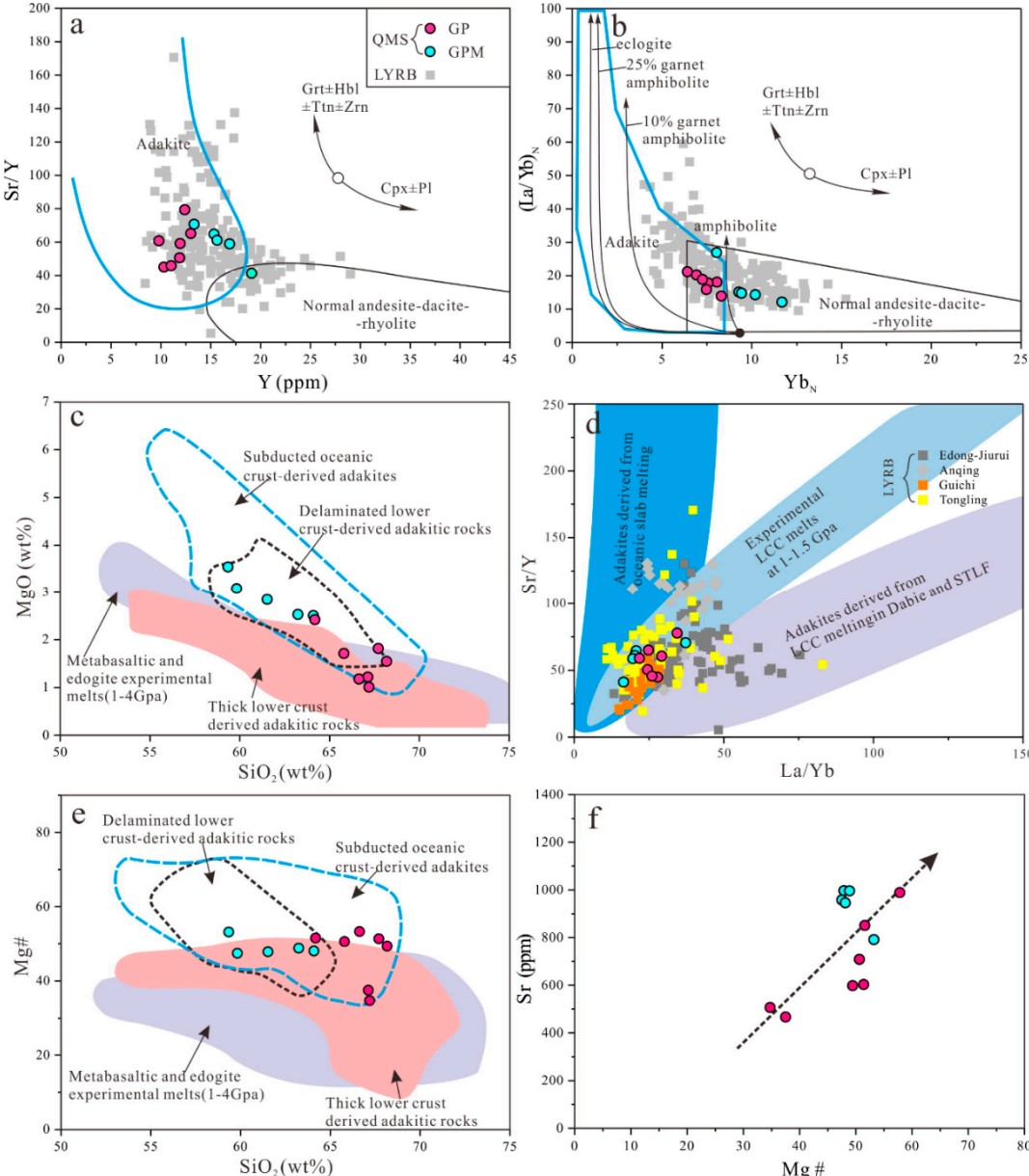

**Figure 7.** (**a**) Sr/Y–Y (after [31]), (**b**) Sr/Y vs. (La/Yb)$_N$ classification diagram (after [31]), (**c**) MgO vs. SiO$_2$, (**d**) Sr/Y vs. La/Yb (after [32]), (**e**) Mg# vs. SiO$_2$, and (**f**) Mg# vs. Sr for the Qiaomaishan intrusive rocks and other MLYRB adakitic rocks. The area of subducted oceanic adakites is from [9,17], the area of experimental lower continental crust (LCC) melts (1–4 GPa) is from [33], the area of delaminated lower crust-derived adakitic rocks is from [34,35], and the area of thick lower crust-derived adakitic rocks is from [36]. Data for MLYRB adakitic rocks are from [9,16,32,37].

The REEs of the Qiaomaishan intrusive rocks exhibit strong enrichments in LREEs relative to HREEs as well as different LREE/HREE ratios (11.9–15.2 for GP, and 9.9–14.9 for GPM, Table S1)

with slightly negative Eu anomalies (0.86–0.93). GP has systematically lower REE contents and more depleted HREE than GPM (Figure 8a). The intrusive rocks share similar normalized trace element patterns (Figure 8b) characterized by an enrichment of large ion lithophile elements (LILEs; e.g., Ba, Rb, and Th), and depletion of high field strength elements (HFSEs) with positive anomalies of Sr and Pb and pronounced negative anomalies of Ta, Nb, and Ti (Figure 8b). However, GPM has systematically higher Cr (27.6–40.5 ppm), Ni (15.84–31.39), and V (99.4–138.0) contents than GP (Cr: 7.6–21.5 ppm, Ni: 4.48–20.12 ppm, and V: 55.9–77.7).

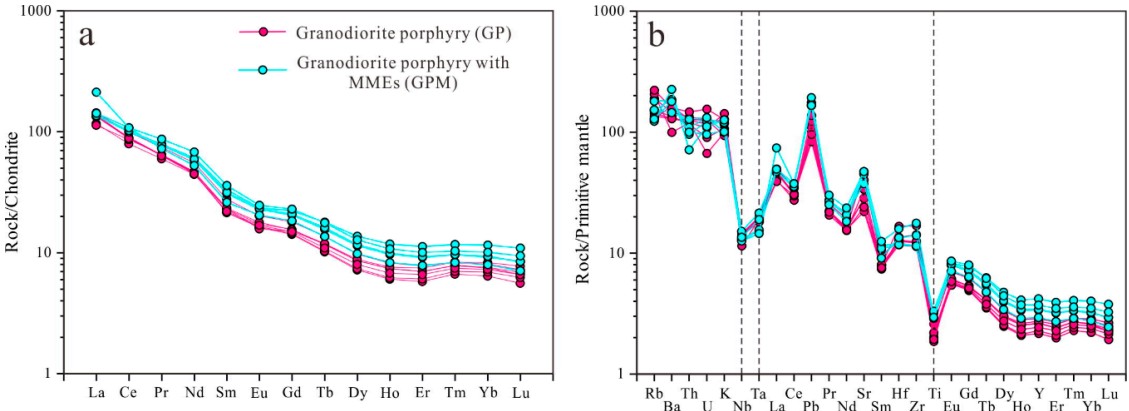

**Figure 8.** (**a**) Chondrite-normalized REEs and (**b**) the primitive mantle-normalized trace element distribution patterns of the Qiaomaishan intrusions. Chondrite- and primitive mantle-normalized data are taken from [38].

### 4.2. Mineral Chemistry and Isotopic Studies

The microprobe analyses and calculated results of plagioclase crystals are listed in Supplementary Table S2. Plagioclases exhibit complex compositional zoning and a reverse zonal texture (Figure 4a,b). Plagioclases with Ab-rich (68–70) cores surrounded by overgrowths of An-rich (25–32) rims show boundaries between the rim and the core (Figure 4a,b). Furthermore, the An values of plagioclase show a positive correlation with $Ca^{2+}$ (also $Fe_2O_3{}^T$ and MgO). Thus, the Ca, Fe or Mg contents of the plagioclase abruptly increase from core to rim, indicating that new magma was injected into the magma chamber [30].

Zircon LA–ICP–MS U–Pb data and the trace element results are listed in Supplementary Tables S3 and S4. Most of the zircon grains from GP and GPM are euhedral and distinctly characterized by oscillatory zoning (Figure 9c,d), with high Th/U ratios ranging from 0.35 to 1.31 (Table S3), which implies magmatic origins. Zircons from GP and GPM yielded weighted mean $^{206}Pb/^{238}U$ ages ranging from 135.1 ± 2.0 Ma (*n* = 13, MSWD = 0.85) to 134.9 ± 1.6 Ma (*n* = 23, MSWD = 0.79), suggesting Early Cretaceous formation ages. Although GPM is compositionally different due to its MME component, it shares similar U–Pb ages (Figure 9a,b) and zircon trace elements patterns (Figure 9c,d) to those of its host adakites (GP), implying that the two types of rock units originated from coexisting magmas.

All the zircons from the Qiaomaishan intrusive rocks display depleted LREE and enriched HREE distribution patterns, with pronounced positive Ce anomalies and considerable negative Eu anomalies, suggesting that they belong to magmatic zircons. Ti-in-zircon temperatures were calculated in [39], which yielded crystallization temperatures of 681–950 °C. The calculated zircons Eu/Eu* and $Ce^{4+}/Ce^{3+}$ ratios [40,41] of GP and GPM share homogeneous variations, with Eu/Eu* ratios from 0.37–0.72 and 0.46–0.74, and $Ce^{4+}/Ce^{3+}$ ratios 175 to 525 and 150 to 683 for GP and GPM, respectively, similar to the adakitic relationship to Cu–Au mineralization in the Middle–Lower Yangtze River Belt (MLYRB) [9,16,32,37].

In situ Hf isotopic results for the zircons are listed in Table S5. The calculated zircon $\varepsilon_{Hf}(t)$ values (t = 135 Ma) for GP and GPM vary from −6.7 to −12.2 and −5.7 to −10.6, with corresponding two-stage Hf model ages ranging from 1.5 to 2.0 Ga and from 1.5 to 1.9 Ga, respectively (Figure 10a,b).

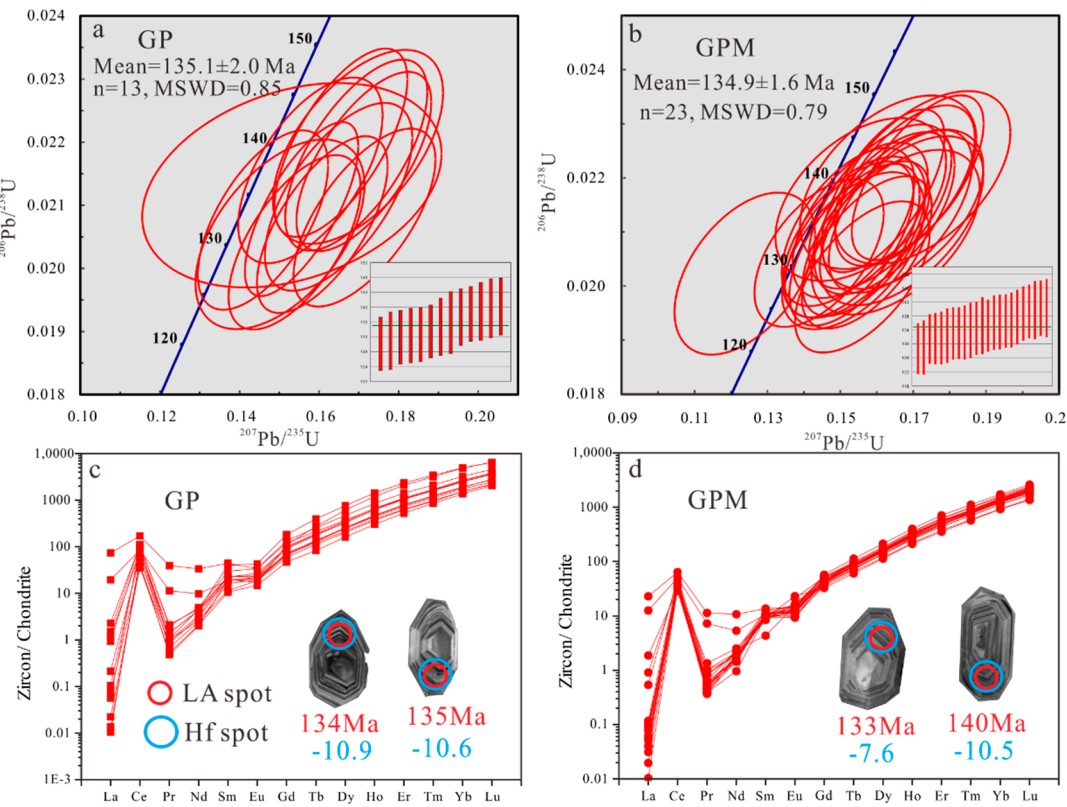

**Figure 9.** (**a**) Zircon U–Pb Concordia diagrams and (**b**) weighted average diagrams of the intrusions in Qiaomaishan. (**c**) representative CL images and (**d**) REE distribution patterns of zircons in the Qiaomaishan intrusions. LA (U–Pb and REE) spots: red circles; Hf spots: blue circles.

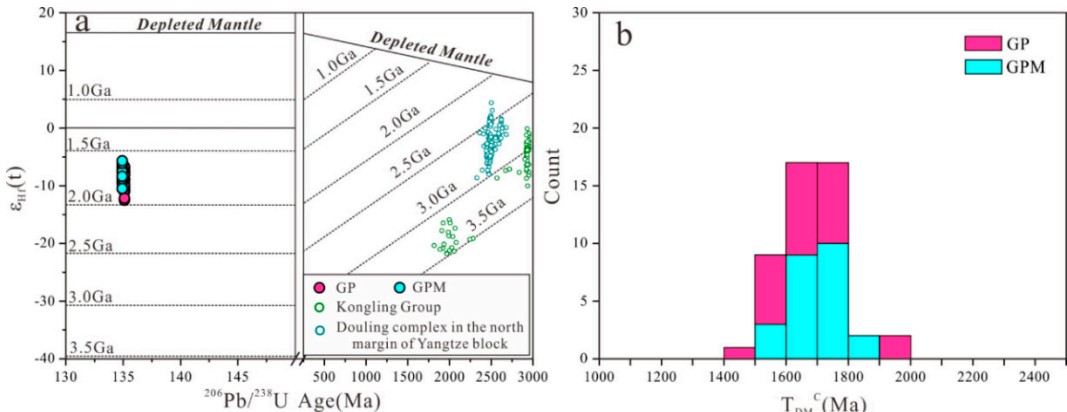

**Figure 10.** (**a**) Zircon Lu–Hf isotopic compositions and (**b**) $T_{DM}2$ model ages of the Qiaomaishan intrusions. Data for the Kongling Group are from [42]. Data for the Douling complex are from [43].

### 4.3. Sr–Nd–Pb Isotopes

The Sr–Nd–Pb isotopic data for the Qiaomaishan intrusive rocks are shown in Supplementary Table S6. The initial Sr–Nd isotopes of GP and GPM exhibit nearly the same range, with high $(^{87}Sr/^{86}Sr)_i$ from 0.70602 to 0.70667 and low $\varepsilon_{Nd}(t)$ values of −6.4 to −9.1 (Figure 11), which are different from the

values of the granodiorites associated with the W–Cu deposit in the Jiangnan Tungsten belt [10,44]. In addition, they have high radiogenic Pb isotopic compositions, with $(^{206}Pb/^{204}Pb)_i = 18.41$–18.86, $(^{207}Pb/^{204}Pb)_i = 15.62$–15.67 and $(^{208}Pb/^{204}Pb)_i = 38.64$–39.16 (Figure 12a,b), which are plotted in the field of MORB [45] between Early Cretaceous mafic rocks [16,34,37] and EM II end-members. However, the Sr–Nd–Pb isotopic compositions of GPM result in more depletion than those of GP (Figures 11 and 12), coming close to the area of the Early Cretaceous mafic rocks in MLYRB.

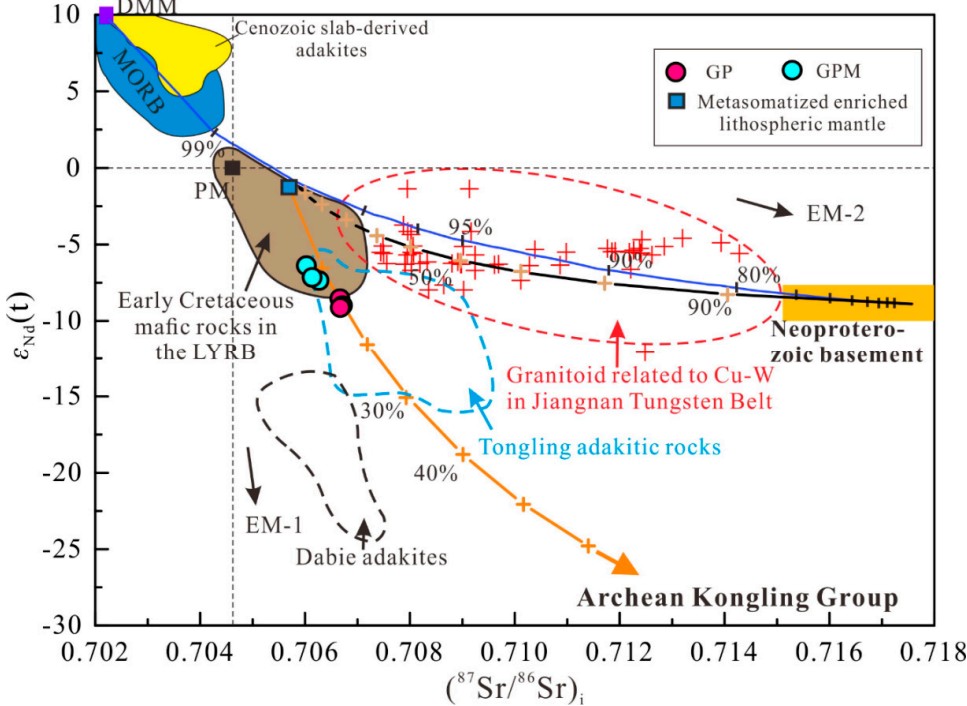

**Figure 11.** Initial Sr–Nd isotopic compositions of the Qiaomaishan intrusions. Note: End-members used for binary mixing calculated lines are depleted mantle (DMM) with Sr = 21 ppm, Nd = 0.4 ppm, and $^{87}Sr/^{86}Sr(t) = 0.7022$; $\varepsilon_{Nd}(t) = 10$ [46]. The metamorphic enriched lithospheric mantle is defined by the average values of the NNS formation volcanic rocks [37]. Archean Kongling Group with Sr = 315 ppm, Nd = 39 ppm, $^{87}Sr/^{86}Sr(t) = 0.7177$, and $\varepsilon_{Nd}(t) = -36.4$ [47,48]. Data source: Dabie adakites [32], Tongling adakitic rocks [9,16], Early Cretaceous mafic rocks [16,34,37], granodiorites from Jiangnan W belt [10,44,49], PM, EM-1, and EM-2 [45].

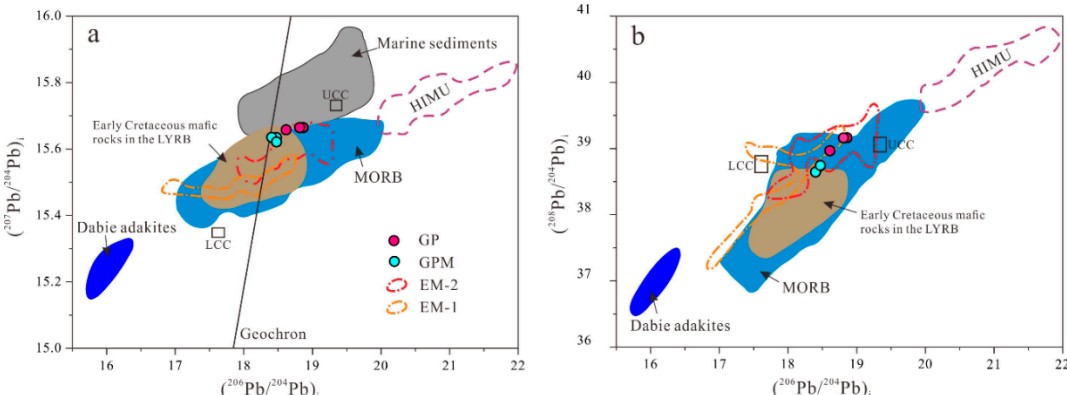

**Figure 12.** Initial Pb isotopes. (**a**) $(^{207}Pb/^{204}Pb)_i/(^{206}Pb/^{204}Pb)_i$ and (**b**) $(^{208}Pb/^{204}Pb)_i/(^{206}Pb/^{204}Pb)_i)$ ratios of the Qiaomaishan intrusions. Data sources for MORB, Marine sediments, HIMU, EM-1, and EM-2 are from [45]; Early Cretaceous mafic rocks in the MLYRB are based on [16,34,37] and Dabie adakites [32].

## 5. Discussion and Conclusions

*5.1. Magma Origin*

As mentioned above, the Early Cretaceous formation ages (135.1 ± 2.0 Ma, 134.9 ± 1.6 Ma) of the ore-bearing granodiorite porphyries in the Qiaomaishan are consistent with the first episode of mineralization in the MLYRB. These porphyries belong to a high-K calc-alkaline series with high Mg# (37–53) and are distinguished by fractionated LREE/HREE and enriched LILEs (e.g., Rb, Ba, and Sr) without Eu anomalies. In particular, they show high Sr compositions (466–988 ppm), Sr/Y (44.99–79.41), and $(La/Yb)_N$ values (14–27), as well as enriched Hf–Sr–Nd isotopic compositions ($\varepsilon_{Hf}(t)$ = −6.7 to −12.2, $I_{Sr}$ = 0.70602–0.70667, and $\varepsilon_{Nd}(t)$ = −6.4 to −9.1), indicating adakite affinities. In adakite discrimination diagrams (Figure 7c–e), samples of the granodiorite porphyry lay between the areas of crust-derived adakites and oceanic crust-derived adakites.

The origin of magma and petrogenesis of adakites related with the first episode of Cu polymetallic mineralization in the MLYRB has been studied extensively and still remains controversial. Major models include (1) partial melting of the subducted Paleo-Pacific oceanic crust with the involvement of enriched compositions [9,16,32,50]; (2) melting of delaminated or thickened mafic lower crust [34,35,51]; (3) fractional crystallization (FC) of basaltic magmas, possibly followed by contamination of the continental crust [52–54]; and (4) magma mixing of mantle-derived mafic and crust-derived felsic melt [30,37,55,56].

(1) Although the enriched Sr–Nd isotopic characteristics of the Qiaomaishan adakites can be obtained by adding enriched oceanic sediments or lower crust [9,57], more than 45–70% of the enriched composition is required [30]. Such a large addition would have obviously modified the geochemical features of these rocks, e.g., given them a high $SiO_2$ content, which is inconsistent with low $SiO_2$ (58–65 wt. %) observed in the Qiaomaishan intermediate rocks. Furthermore, ore-bearing adakites have a positive correlation between Sr/Y and $(La/Yb)_N$ ratios (Figure 6d) and were plotted in the LCC melt region, suggesting a crustal adakite nature, different from the trend observed for the arc adakites derived from the subducted oceanic crust. Additionally, this model fails to explain the zircon O isotopic compositions of ore-bearing adakites in MLYRB [53,58]. The model of partial melting of subducted oceanic crust is not suitable for ore-bearing adakites in Qiaomaishan.

(2) Based on enriched isotopic compositions, scholars proposed a model of delaminated or thickened mafic lower crust [34,51]. Rapp and Watson [59] proposed that adakites were directly originated from partial melting of the mafic lower continental crust with low Mg# (<40) in contrast with the high Mg adakites from Qiaomaishan. Even though the melts derived from subsequently delaminated LCC would have acquired a high Mg# signature via their interaction with the peridotite mantle, this factor cannot explain the existence of the widespread mafic enclaves and textural disequilibrium (Figure 4) observed in the Qiaomaishan granodiorite porphyry. These petrological characteristics are attributed to the injections of fresh magma into the crustal magma chamber [30,60]. Indeed, the experimental modeling studies of Qian and Hermann [33] and Ma et al. [61] confirmed that the "adakitic" geochemical signatures of rocks in the residue melts can be inherited from the melted LCC without an over thickened or delaminated lower continental crust at mantle depths. Therefore, the high Mg# adakites in Qiaomaishan are not directly formed by the partial melting of a delaminated or thickened mafic lower crust.

(3) In another model, adakite signatures can be obtained by the assimilation–fractional crystallization of basaltic magmas [52,62]. In major element Harker diagrams (Figure 6), CaO, MgO, $P_2O_5$, $Fe_2O_3^T$, and $TiO_2$ display a negative correlation with an increase in $SiO_2$, indicating that the ore-bearing adakites were likely affected by fractional crystallization. However, the lack of a significant correlation between silica contents and $Al_2O_3$, $K_2O$, and $Na_2O$ (Figures 5b and 6) suggests that the FC processes produced only a limited amount of silica, and the Qiaomaishan intrusions could not have originated from the fractional crystallization of a single basic magma. Notably, Qiaomaishan adakites have a negative correlation between $SiO_2$ with Y (Figure 6g), which is inconsistent with the



condition that the Y contents in a melt will increase as basaltic to andesitic magma suffers olivine and pyroxene FC processes [62]. Furthermore, there are no correlations between $SiO_2$ with Sr/Y and La/Yb (Figure 6h,i). Importantly, the $\varepsilon_{Nd}(t)$ and $I_{Sr}$ of the Qiaomaishan adakites also exhibit correlative variation with silica (Figure 13) and display trends of magma mingling or assimilation in the $I_{Sr}$ and $\varepsilon_{Nd}(t)$ versus 1/Sr and 1/Nd diagrams (Figure 13), implying that a simple FC model is not favorable. Furthermore, Xu et al. [51] argue against the AFC model, suggesting that there is a small quantity of mafic intrusive rocks coexisting with the adakites. Large-scale assimilation would also significantly change the Sr–Nd isotopic composition, which is inconsistent with the small variations of $I_{Sr}$ from 0.70602 to 0.70667 and of $\varepsilon_{Nd}(t)$ from −6.4 to −9.1 in Qiaomaishan, thereby indicate of little assimilation. In this regard, assimilation of the crust did not play a significant role in the magma process. Therefore, we suggest that the AFC model is not feasible.

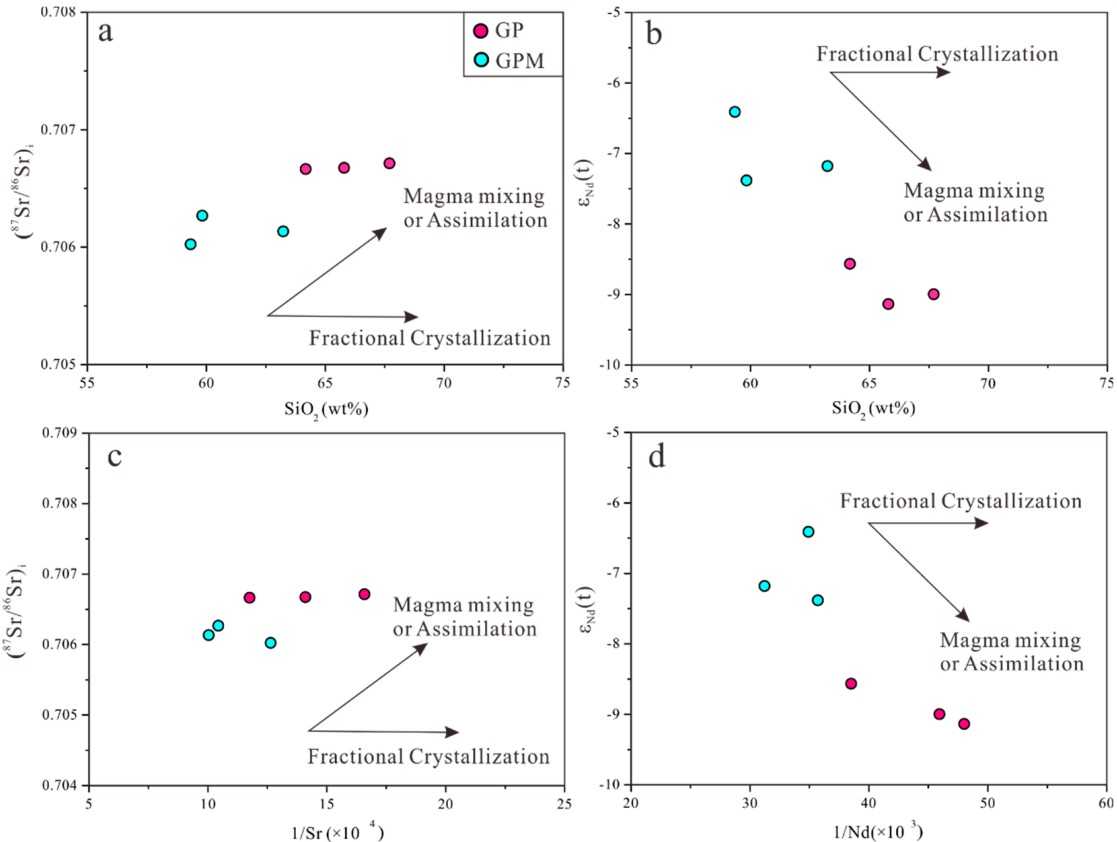

**Figure 13.** Plot of (**a**) $^{87}Sr/^{86}Sr$ versus $SiO_2$ (wt. %), (**b**) $\varepsilon_{Nd}(t)$ versus $SiO_2$ (wt. %), (**c**) $\varepsilon_{Nd}(t)$ versus 1/Sr, and (**d**) $\varepsilon_{Nd}(t)$ versus 1/Nd of the Qiaomaishan intrusions.

(4) Consequently, our study focuses on the magma mixing model as the most feasible interpretation for the genesis of the ore-bearing granodiorite porphyry, as well as regional Cu–W mineralization. This model is supported by the following views:

(a) Mafic enclaves in the granodiorite porphyry commonly have faded contacts with the host rocks (Figure 3c) and are without chilled margins. Coupled with their similar zircon U–Pb ages, this indicates that the granodiorite porphyry and the mafic enclaves coexisted as contemporaneous magmas. Notably, there are a large number of mineral disequilibrium textures in the Qiaomaishan, e.g., corrosion texture of quartz (Figure 4c), dark minerals with microcrystalline textures (Figure 4d), and significantly diablastic textures of the hornblendes (Figure 4e). In addition, mafic enclaves contain abundant minor acicular apatites and fine-grained hornblendes (Figure 4f), suggesting the quenching process of the mafic magmas injected into [30,60] the felsic magmas.

(b) The reverse zonal and disequilibrium texture of the plagioclase. The plagioclase from the granodiorite porphyry displays widespread compositional zonation (Figure 4a), a variable An value (Figure 4b), and $Ca^{2+}$ contents between the core and rim (Figure 4b), which suggest that new magma was injected into the magma chamber [30].

(c) The intrusive rocks of the Qiaomaishan are marked by high Mg# (37–53), like most of the adakitic rocks in MLYRB, and these values are higher than those of rocks derived from experimental LCC melts (Figure 7e). Thus, high Mg# mafic magmas are required in the source, which is consistent with the evidence that GPM has higher Mg# than the host rocks. This is explained by the geochemical compositions that were modified after mafic magmas were injected into the felsic magma [30]. Furthermore, the positive correlation of Sr contents and Mg# in the Qiaomaishan intrusive rocks suggests that high LILE elements (e.g., Sr) were feasibly derived from the mafic magma formed by an enriched mantle [37] that was metasomatized [57] by the melts/fluids [9,16,32,50] released from subduction slab.

(d) Zircon from GP and GPM yielded similar U–Pb ages, which show slight variations in their $\varepsilon_{Hf}(t)$ values (–6.7 to –12.2 and –5.7 to –10.6, respectively), coupled with large variations in their Nd isotopic and whole-rock compositions; these are strong indications of magma mixing between felsic and mafic magmas during the magma evolution processes.

## 5.2. Source Characteristics

As shown above, there are large differences in the geochemical compositions (e.g., high Mg#, Cr, Ni, and V contents, and enriched LILEs) between the granodiorite porphyry with MMEs and granodiorite porphyry, due to the presence of mafic enclaves. The positive correlation between Sr contents and Mg# (Figure 7f) implies that mafic magma is derived from an enriched mantle and not form thickened LCC. Moreover, this condition is supported by the Sr–Nd–Hf isotopic compositions of the Qiaomaishan intrusive rocks. GPM has more depleted $I_{Sr}$, $\varepsilon_{Nd}(t)$ and $\varepsilon_{Hf}(t)$ values than the GP (Figures 11 and 12) that appeared in the source area, overlapping with regions of Early Cretaceous mafic rocks in the MLYRB derived from the enriched lithospheric mantle [37] and metasomatized by subduction slab melts/fluids [9,16,32,50], as suggested by studies [57,63]. Collectively, these observations suggest that mafic magma is derived from mantle.

The granodiorite porphyry has an abundance of euhedral hornblende crystals as well as MMEs, which reveals that the parental magma was hydrous. In addition, the calculated zircon $Ce^{4+}/Ce^{3+}$ and Eu/Eu* ratios qualitatively constrain the magmatic oxygen fugacity [40,41]. Our results suggest that Qiaomaishan adakites have high $Ce^{4+}/Ce^{3+}$ and Eu/Eu* ratios (Figure 14a), coupled with high log $fO_2$ located above the FMQ buffer, as shown in Figure 14b; these ratios are similar to those of other Cu–Au-bearing adakites in the MLYRB and clearly higher than those of the granitoids related to the W–Cu deposits in the Jiangnan Tungsten Belt (Figure 14a), indicating the high oxygen fugacity in the magma source. Importantly, it is widely accepted that the mantle wedges [64] metasomatized by the fluids/melts produced by subducted plates [65] have high oxygen fugacity [66–69]. Furthermore, in the Lower Yangtze River area, there are widespread mafic magmas underplated at LCC depth as revealed by deep seismic reflection profiles [70]. Coupled with depleted HFSEs, enriched LILEs (Figure 8a), and significantly negative Nb–Ta–Ti anomalies (Figure 8b), mafic magma with high magmatic water content and oxygen fugacity originated from the enriched lithospheric mantle that had been metasomatized by melts/fluids released from subduction slab.

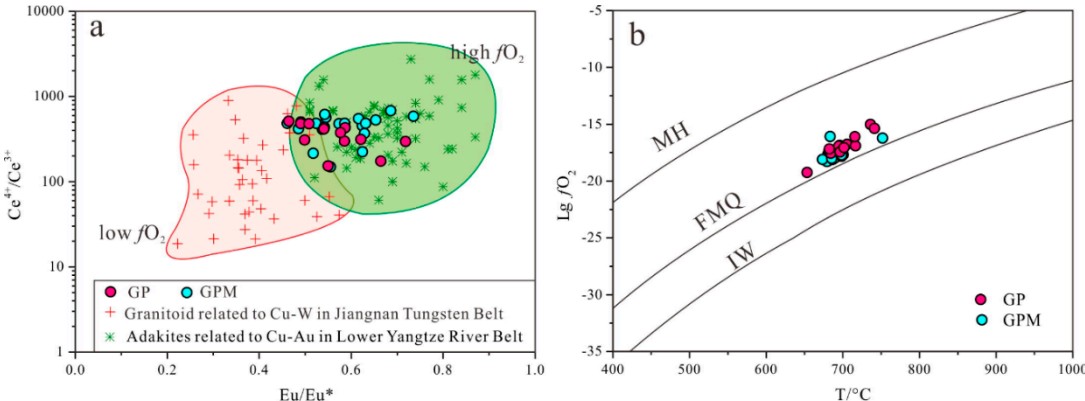

**Figure 14.** Plot of (**a**) $Ce^{4+}/Ce^{3+}$ value vs. Eu/Eu* value and (**b**) log $fO_2$ vs. T (°C) of zircon samples from the Qiaomaishan intrusions. Zircon $Ce^{4+}/Ce^{3+}$ and Eu/Eu* values are calculated using the method in [40]. MH: magnetite–hematite buffer, FMQ: fayalite–magnetite–quartz buffer, IW: iron–wüstite buffer. Data sources: adakites related to Cu–Au in the MLYRB [9,57], granitoid related to Cu–W in Jiangnan Tungsten Belt [12,49].

The enriched isotopic features of these rocks indicate that a large amount of crust components are involved in the source region. Therefore, we propose that felsic end-member magma could be generated from the LCC in the MLYRB. The existence of "Kongling–Dongling" Precambrian basements consisting of mafic granulites/amphibolites [71] and Archean TTGs in the MLYRB were demonstrated by several studies [1,25,72]. The Qiaomaishan intrusive rocks exhibit high $Na_2O$ contents (3.80–4.55) and $Na_2O/K_2O$ ratios (0.94–1.50); thus, we propose that the felsic melts were formed by the melting of basement components with Na-rich mafic granulites/amphibolites [59,73]. Consequently, we suggest that felsic magma is derived from the Precambrian mafic granulites/amphibolites crustal component.

In summary, the Qiaomaishan granodiorite porphyries are the product of the mixing of a felsic magma derived from the partial melting of the lower continental crust and a mafic magma generated from the metasomatized enriched lithosphere mantle.

### 5.3. Source of Cu and W

High oxygen fugacity and fluid- and Cu-rich source regions are favorable for the formation of copper polymetallic deposits [66,68,74–76]. In general, Cu is incompatible in most of the common silicate phases [74,77]. Cu's behavior is controlled by sulfide, whereas the behavior of sulfur is controlled by oxygen fugacity [78]. Under a high oxidation state, sulfides are destabilized, and Cu is efficiently transported into the silicate melt [79]. The process of magma mixing results in an increase in volatile contents [80], inhibiting the separation of sulfides in the melt and releasing $SO_2$ accompanied by the transport of Cu [74,79], resulting in Cu enrichment. Sulfate reduction and oxygen fugacity variations eventually lead to Cu mineralization [79], which is ascribed to magnetite crystallization [81]. Slab-derived fluids/melts have high $H_2O$ and oxygen fugacity [81,82]. These properties are inherited in mantle wedges metasomatized by fluids/melts produced by subducted plates. Therefore, chalcophile elements (Cu, S) are feasibly derived from the enriched lithospheric mantle that has been metasomatized by fluids/melts released from the subduction slab.

As discussed above, Cu mainly comes from the mantle source, whereas the concentration of W in the mantle is 0.13 ppm—much lower than that in the crust (the upper crust is 2.0 ppm, and it is 0.7 ppm in the lower crust) [83], indicating that W is mainly derived from crust [84,85]. Liu et al. [86] proposed that Cu–W deposits can be explained by superimposed mineralization caused by two stages of magmatic activity, whereas no other magmatic activity was observed in Qiaomaishan [8]. Most plausibly, the formation of Cu–W deposits is controlled by the source of intrusive rocks, and most of them are part of the crust–mantle mixed domain [12,49,87]. In particular, some researchers [88–90]

recently proposed that Dongling basement materials are likely the main source of tungsten in the MLYRB, based on the Nd–Sr isotopic compositions of scheelite. As mentioned above, the assimilation of Qiaomaishan intrusive rocks is not very significant and is mainly controlled by the mixing of mafic magmas derived from mantle and felsic melts derived from the crust. Therefore, we propose a situation where the tungsten in the Qiaomaishan deposit was derived from the partial melting of an LCC source heated by the enriched lithospheric mantle.

*5.4. Ore-Forming Process*

During the Late Mesozoic, the Lower Yangtze area was an active continental margin setting [91] and was closely associated with the subduction of the Paleo-Pacific oceanic plate [92], resulting in the widespread emplacement of Late Mesozoic intrusive rocks [93–95]. Subduction of the Paleo-Pacific slab resulted in intense dehydration, and the overlying mantle wedge was metasomatized by the melts/fluids, forming an enriched lithospheric mantle. The slab-released fluids caused the melting of the overlying mantle wedge, generating mafic magmas with a large amount of Cu content as well as volatiles and $H_2O$ in a high oxygen fugacity environment. Mafic magmas provide heat and $H_2O$, further inducing the partial melting of the LCC and generating W-rich felsic magmas. Subsequently, mafic and felsic magmas rapidly ascended and mixed in the shallow magma reservoir. Due to the diversity of those magmas, they could not achieve complete homogenization. Subsequently, mixed magmas with abundant copper and tungsten metals were emplaced at shallow levels and produced granodiorite porphyries with mafic enclaves. Hydrothermal processes then generated Cu mineralization during the early skarn stage. However, W is relatively inactive in high oxygen fugacity hydrothermal fluids [96]. As oxygen fugacity became reduced in the hydrothermal fluid [49,97,98], tungsten precipitated and was deposited as scheelite in the late tungsten mineralization stage and carbonate stage.

**Supplementary Materials:** The following are available online at http://www.mdpi.com/2075-163X/10/2/171/s1, Table S1: Major (wt. %) and trace element (ppm) results of the Qiaomaishan intrusive rocks; Table S2: Major compositions of plagioclase from the granodiorite porphyry in Qiaomaishan; Table S3: Zircon LA-ICP-MS U-Pb analytical data from the Qiaomaishan intrusive rocks; Table S4: Zircon trace elements and $Ce^{4+}/Ce^{3+}$ ratios and $T_{\text{Ti in Zircon}}$ of the Qiaomaishan intrusive rocks; Table S5: MC-ICP-MS zircon Lu-Hf isotopic compositions of the Qiaomaishan intrusive rocks; Table S6: Whole-rock Sr–Nd–Pb isotopic compositions of the Qiaomaishan intrusive rocks.

**Author Contributions:** H.Q. wrote the paper; S.L., X.Y., and J.D. designed the experiments; Y.Z., L.Z., and J.L. took part in the field investigation; I.L. revised the manuscript. All authors have read and agreed to the published version of the manuscript.

**Funding:** This research was supported by the National Key Research and Development Program of China (2016YFC0600209), Natural Science Foundation of China (41673040), and Project of Geological Science and Technology of Anhui Province (2016-K-4).

**Acknowledgments:** The authors are grateful to two anonymous reviewers for their helpful comments and suggestions that greatly helped to improve an earlier manuscript version. Also, we appreciate for S.L. Qian, Z.J. Xie, Z.F. Yu from No.322 Unit of Bureau of Geology and Mineral Exploration of Anhui Province for field assistance.

**Conflicts of Interest:** The authors declare no conflict of interest.

## Appendix A. Analytical Methods

Zircon grains from two samples (GP, GPM) were first separated through crushing samples to about 60 mesh, sieving in water, heavy liquid and magnetic separation. Based on size, clarity, color, and morphology of zircons under a binocular microscope, single zircon crystals were handpicked and mounted on double-sided sticky tape prior to casting in an epoxy mount and burnished down to near crystal centers. Prior to U–Th–Pb and Lu–Hf isotope analyses, optical microscopy and cathodoluminescence studies were performed to reveal internal structures and zoning at Analytical Center of the University of Science and Technology of China (USTC).

*Appendix A.1. Zircon LA-ICM-MS U–Pb Dating*

Zircon U–Pb dating and trace element analyses of zircons were conducted synchronously using LA-ICP-MS were conducted at the School of Resources and Environmental Engineering, Hefei University of Technology. The laser-ablation system was a GeoLas 200 M equipped with a 193 nm ArF-excimer laser, operated at a constant energy of 80 mJ, with a repetition rate of 8 Hz and a spot diameter of 31 μm. Data were acquired for 30 s with the laser off, and 40 s with the laser on, giving approximately 100 mass scans. Helium was used as carrier gas sampling ablation aerosols to the ICP source for Analysis. NIST SRM 610 glass and Temora zircons were used as external calibration standards and 29Si as the internal standard. Off-line selection and integration of background and analyte signals, and time-drift correction and quantitative calibration for trace element analyses and U–Pb dating were performed by ICPMSDataCal v.10.8 [99]. Concordia and weighted mean U–Pb ages were generated using Isoplot v. 3.3 at the 2σ uncertainty [100].

*Appendix A.2. In-Situ Zircon Lu–Hf Isotopes*

In-situ analyses of Lu–Hf isotopes was conducted on a Nu Plasma II MC–ICP–MS connected to a RESOLution M-50 193 nm laser system, at the State Key Laboratory of Continental Dynamics in Northwest University. The detailed instrumental parameters were described by [101]. The laser-ablated spot was measured on the MC–ICP–MS instrument with a beam diameter of was 44 μm, while the laser repetition rate was 6 Hz and the energy density applied was 6 J/cm$^2$. The mass fractionations of Hf and Yb were calculated with reported 0.7325 for $^{179}$Hf/$^{177}$Hf and 1.1248 for $^{173}$Yb/$^{171}$Yb. All the Lu–Hf isotope results are reported in 2σ error. The recommended $^{176}$Hf/$^{177}$Hf ratio of standard zircon 91500 is 0.282289 ± 0.000011. The obtained Hf isotopic compositions are 0.282015 ± 0.000009 (2σ, *n* = 4) for GJ-1, 0.282787 ± 0.000007 (2σ, *n* = 4) for Monastery, respectively. In the calculation of $\varepsilon_{Hf}$(t), the recommended decay constant value of $^{176}$Lu is $1.867 \times 10^{-11}$ yr$^{-1}$, the $^{176}$Lu/$^{177}$Hf and $^{176}$Hf/$^{177}$Hf values of chondrite are 0.0336 and 0.282785, respectively [102]. Single-stage Hf model ages (TDM1) were calculated with respect to the depleted mantle with a present-day $^{176}$Hf/$^{177}$Hf ratio of 0.28325 and a $^{176}$Lu/$^{177}$Hf ratio of 0.0384 [103]. Two-stage Hf model ages were calculated by projecting the initial $^{176}$Lu/$^{177}$Hf values of zircon back to the depleted mantle growth curve using $^{176}$Lu/$^{177}$Hf = 0.015 for the average continental crust [104].

*Appendix A.3. Microprobe Analyses*

Mineral Microprobe analyses were measured at the School of Resources and Environmental Engineering, Hefei University of Technology by a JEOL JXA-8230M electron microprobe, using an accelerating voltage of 15 kV, a probe current of 20 nA, and a beam diameter of 5 μm.

*Appendix A.4. Major and Trace Elements*

T Whole-rock major and trace elements were analyzed at the Ministry of Land and Resources P.R.C. Hefei Mineral Resources Supervision and Testing Center. Samples were powdered using an agate mill to grain seizes b200 mesh. Major elements were determined using XRF spectrometry with standard deviations within 5%. The detailed methodology is as follows: Loss of ignition (LOI) was determined after igniting sample powders at 1000 °C for 1 h. A calcined or ignited sample (0.9 g) was added with lithium borate flux (~9.0 g, 50% $Li_2B_4O_7$–$LiBO_2$), mixed well and fused in an auto fluxer at a temperature between 1050–1100 °C. A flat molten glass disk was prepared from the resulting melt. This disk was then analyzed by wavelength-dispersive X-ray fluorescence spectrometry (XRF) using an AXIOS Minerals spectrometer. Trace elements, including REE, were determined by inductively coupled plasma mass spectrometry (ICP–MS) of solutions on an Elan DRC-II instrument (Element, Finnigan MAT) after 2-day closed beaker digestion using a mixture of HF and $HNO_3$ acids in Teflon screw-cap bombs. Detection limits, defined as 3 s of the procedural blank, for some critical elements are as follows (ppm): Th (0.05), Nb (0.2), Hf (0.2), Zr (2.2), La (0.5) and Ce (0.5). Accuracy and precision

of the data are better than 5% for major elements and 10% for trace elements on the basis of analytical results and replicate analyses of international standard reference material (SRM).

*Appendix A.5. Whole-Rock Sr–Nd–Pb Isotopes*

All Sr–Nd–Pb isotopes were isolated at the CAS Key Laboratory of Crust-Mantle Materials and Environments, USTC. For Sr–Nd isotope determination, sample powders were initially dissolved in Teflon capsules with $HClO_4$ +HF solution for ~7 days at ~120 °C. Then the solution was dried and redissolved by HBr acid. For lead separation, the solution was purified twice through AG1-X8 resin. Rb, Sr and light rare-earth elements were isolated on quartz columns by conventional ion-exchange chromatography with a 5-ml resin bed of AG 50W-X12 (200–400 mesh). Nd and Sm were separated from other rare earth elements on quartz columns using1.7 ml Teflon powder coated with HDEHP, di (2-ethylhexyl) orthophosphoric acid, as a cation exchange medium. The lead was purified using a conventional anion-exchange method using HBr as an eluant. More details on analytical procedures are given in Chen, et al. [105]. Sr–Nd–Pb isotope ratios were determined on a Finnigan MAT–262 mass spectrometer in USTC. Sr and Nd isotopic ratios were normalized to 0.1194 for $^{86}Sr/^{88}Sr$ and 0.7219 for $^{146}Nd/^{144}Nd$. NBS981 standard solution was repeatedly measured for mass fractionation correction on the measured Pb isotopic ratios of samples. NBS987 and La Jolla standard solutions analyzed along with samples yielded 0.710250 ± 12 (2σ) for $^{87}Sr/^{86}Sr$ and 0.511860 ± 12 (2σ) for $^{143}Nd/^{144}Nd$. The precision of the measured Sr and Nd isotopic ratios is better than 0.003% and Pb isotopic ratios better than 0.01%.

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
