# Peer review of "The Role of Magma Mixing in Generating Granodioritic Intrusions Related to Cu–W Mineralization: A Case Study from Qiaomaishan Deposit, Eastern China"

_minerals, doi:10.3390/min10020171_

Round 1

Reviewer 1 Report

I appreciated the paper, structured on a robust petrological basis with the aim of explaining the peculiar metallogenesis of a Cu-W deposit of the investigated district. The various analytical methods used have provided broad support for the discussion, which in general seems to me well structured, and for the conclusions. However, one point that still penalizes the work is the English language and style, which still needs to be extensively revised: the work is full of small mistakes, typos and several sentences must be reformulated. See the attached text for my detailed notes.

Author Response

Response to Reviewer 1 Comments

Comments: I appreciated the paper, structured on a robust petrological basis with the aim of explaining the peculiar metallogenesis of a Cu-W deposit of the investigated district. The various analytical methods used have provided broad support for the discussion, which in general seems to me well structured, and for the conclusions. However, one point that still penalizes the work is the English language and style, which still needs to be extensively revised: the work is full of small mistakes, typos and several sentences must be reformulated. See the attached text for my detailed notes.

Response: Thank you for this positive comment. Those comments are all valuable and very helpful for revising and improving our paper, as well as the important guiding significance to our research. We revised the contents and errors carefully and made extensive changes in the manuscript according to the suggestions of the reviewers. We tried our best to improve the English writing and contents of the paper. These changes will not influence the framework of the paper. And here we listed the main changes as follow:

We simplified the title and abstract to make it more consistent with the research. You can find in Line 2, 19-21. We improved the expression and contents in the part “Introduction”, for better understanding. You can find in Line 64-69, 19-21. We have also expanded the sample description and test methods. You can find in Line 129-131, and Appendix 1. Analytical methods of the Qiaomaishan intrusions. We changed the abbreviation of two types intrusive rocks, i.e., Granodiorite porphyry (XC01-GP) and Granodiorite porphyry with MMEs (XC08-GPM) in Qiaomaishan, due to XC01and XC08 are easily confused with sample numbers. Meanwhile, we made some changes on legend and order of the Figures, listed in Fig.3, Fig.4, Fig.5, Fig.7, Fig.8, Fig.9, Fig.10, Fig.11, and Fig.12. In the sections 4.1. Whole-Rock Geochemistry, we added description and explanation of Harker diagrams. You can find in Line 213-219. In the discussion 5.1,we rule out large-scale assimilation model on account of the large-scale assimilation will significantly change its Sr–Nd isotopic composition, inconsistent with the small variations of ISr from 0.70602 to 0.70667, and εNd(t) from −6.4 to − 9.1 in Qiaomaishan, indicating that any amount of assimilation. In this regard, assimilation of crust did not play a significant role in magma process. You can find in Line 337-340. In the conclusion, we re-conclude that 2) Cu was derived from mantle magma source, whereas W was originated from LCC-related source region in this district. Magma mixing between mafic and felsic sources is the main mechanism for the formation of the particular association with Cu-W mineralization. You can find in Line 465-467.

Once again, thank you very much for your comments and suggestions, and revision for our manuscript.

Reviewer 2 Report

Title – something is missing from the first part (The role of magma mixing… reads better) – also, please consider shortening this title

Abstract – define acronyms (MLYRB etc.), what do you mean by ‘middle-scale’? Also, abstracts need to be shortened, it needs to be organized as an abstract (with introduction, problem, objectives, methodology, etc.) and it needs to be written in English (the language used here is really hard to understand – please employ a native English speaker to proof this manuscript).

2 – Cu-S skarn – is S a commodity?

Introduction – needs to be re-worked so the problem and objectives can be better explained; e.g., what does mafic enclaves have to do with all this? Etc. Also, figures are best kept for the Geological Setting section.

Figures 3-4 – to be improved (police size and colors, arrows…)

Section 3 – how many samples? Location?... / refrain using ‘Sample…:’ (write sentences instead). Don’t describe only the samples, but talk about the context (if a granodiorite was sampled, say why – what is the aerial surface occupied by the granodiorite? What is it’s cutting relationship with the mineralization etc.? – what is the context?). It is impossible to know how representative the samples are. The section on zircon should go in a Methodology section.

*Methodology section: missing – the remarks on zircon (p.6) don’t give enough details on the method (instrument used? Standards? QA-QC?), and where is the method used to perform whole rock and isotopic analyses described?

Section 4.1 – ask yourself if all these diagrams are necessary / numbers can be presented in a table to avoid sentences with 10s of numbers. The statements on adakite should be discussed in the discussion section.

Section 4.2. – very short for a section. Define Ab and An (or use albite etc. in the text). Last sentence is an interpretation (a poorly sustained one) that has nothing to do in a Discussion section.

p.9 – Figures should be mentioned in the text before being shown / Was Ti-in-zircon calculated by reference #40? (or was it the #40 method that was used?) / move the interpretations to the Discussion section

Discussion – needs to be completely re-organized. Results should be interpreted first and lead to a model, and ideas should be re-grouped (one section on sources, one of magma mingling etc.)

p.10-11 – a short review of what Adakites are and of there significance in porphyry-style mineralization would be useful. Also, the discussion of whether these are adakite or not should be performed by referring to the many authors that worked on adakites

12 etc. – when you say a model is not suitable, first demonstrate it and explain why you say that 12 – delamination and any other source process have nothing to do with enclaves observed in an upper-crust magma chamber. There are source processes, deep magmatic systems (where most chemical differentiation occur) and the textures acquired late in the upper crust. Please get back to igneous petrology basis before re-writing this section.

Also, the whole section needs to be re-written: start with the data, what does chemistry, isotopes, etc. indicate? Then discuss this in the framework of what we know of adakite etc.

13 – Harker diagrams should be shown and described in the Results section. 14 – (1) please use typical vocabulary – mingling, magma chamber replenishment, … / (2) make sentences / (3) English issue – this section is impossible to understand / (4) not sure I get that one – did zircon formed in the mafic magmas? Or have zircons been exchanged between the felsic and mafic magmas? (what does petro and chemistry suggest?)

Section 5.3.1 – to be re-worked, a lot of things don’t make sense. Cu is incompatible, it will enrich in felsic magma? May be, but there is a lot of Cu in mafic magmas too (replenishing a magma chamber with mafic melt may add a lot of Cu to the system). W should be discussed using new data (how much W is there in the studied rocks?)

16 – diversity??? – it is viscosity contrast that prevents mixing. What do you mean by ‘inert’? Vocabulary to be corrected – you have data on metal sources (not on mineralizing processes)

Conclusion – need to write a conclusion that looks like a conclusion

Author Response

Response to Reviewer 2 Comments

Dear reviewer:

Thank you for this constructive comment. Those comments are all valuable and very helpful for revising and improving our paper, as well as the important guiding significance to our research. We have studied comments carefully and have made correction which we hope meet with approval. Revised portion are marked in red in the paper. The main corrections in the paper and the responds to the comments are as following:

Point 1: Title – something is missing from the first part (The role of magma mixing… reads better) – also, please consider shortening this title

Abstract – define acronyms (MLYRB etc.), what do you mean by ‘middle-scale’? Also, abstracts need to be shortened, it needs to be organized as an abstract (with introduction, problem, objectives, methodology, etc.) and it needs to be written in English (the language used here is really hard to understand – please employ a native English speaker to proof this manuscript).

Response 1: Thank the reviewer for these constructive comments. We simplified the title and abstract to make it more consistent with the research. Title-“Important role of magma mixing in generating the Cretaceous granodioritic intrusions related to the Cu–W mineralization in the Xuancheng ore district, eastern China”. In abstract, MLYRB is the acronyms for “Middle-Lower Yangtze River Metallogenic Belt”, and “middle-scale” was revised as “middle-size”. You can find the detailed revision in the revised manuscript with changes marked (line 2, 19-21).

Point 2: 2 – Cu-S skarn – is S a commodity?

Introduction – needs to be re-worked so the problem and objectives can be better explained; e.g., what does mafic enclaves have to do with all this? Etc. Also, figures are best kept for the Geological Setting section.

Response 2: Thank the reviewer for this valuable comment. Li et al., (2015) described the main ore types are copper ores, sulfur ores, copper-sulfur ores, and iron ores; and pyrite, chalcopyrite, bornite, and magnetite are main ore minerals. They have been regarded as a typical Cu–S skarn deposit before the scheelite was identified in sulfide ores. The abundance of mafic microgranular enclaves (MMEs) in the ore-formation related granodiorite porphyry. We carried out a systematic geochemical and comparative studies of two types of ore-bearing intrusive rocks, i.e., granodiorite porphyry, and granodiorite porphyry with MMEs in Qiaomaishan, to reveal the petrogenesis and possible enrichment mechanism for the Qiaomaishan Cu–W mineralization. Geological Setting section is simplified, and the digital informations are omitted, due to confidentiality requirements for No.322 Unit of Bureau of Geology and Mineral Exploration of Anhui Province.

Reference:

Liu, X.; Duan, L. Geological features and Metallogenic regularity of the Tongshan-Qiaomaishan Cu-S-W-Fe polymetallic ore deposit in Xuancheng City. Geol. Anhui. 2015, 25, 174-178. (In Chinese)

Point 3: Figures 3-4 – to be improved (police size and colors, arrows…)

Response 3: We changed the abbreviation of two types intrusive rocks, i.e., Granodiorite porphyry (XC01-GP) and Granodiorite porphyry with MMEs (XC08-GPM) in Qiaomaishan, due to XC01 and XC08 are easily confused with sample numbers. Meanwhile, we made some changes on legend and order of the Figures, listed in Fig.3, Fig.4, Fig.5, Fig.7, Fig.8, Fig.9, Fig.10, Fig.11, and Fig.12.

Point 4: Section 3 – how many samples? Location?... / refrain using ‘Sample…:’ (write sentences instead). Don’t describe only the samples, but talk about the context (if a granodiorite was sampled, say why – what is the aerial surface occupied by the granodiorite? What is it’s cutting relationship with the mineralization etc.? – what is the context?). It is impossible to know how representative the samples are.

Response 4: Thank the reviewer for these constructive comments. Granodiorite porphyry (GP) and Granodiorite porphyry with MMEs (GPM) ares the main intrusion in this area. They occur in the form of stocks or dikes and emplace into the Carboniferous limestone, which are closely related to the skarn Cu–W mineralization in the contact zone (Figure 2b). We collected a series of granodiorite porphyry samples from the drill cores, including 7 granodiorite porphyries (ZK22+02, –316 to –450 m) and 5 granodiorite porphyries with MMEs (ZK18+02, –107 to –337m), from the contact zone between Carboniferous limestone to the granodiorite porphyry. You can find the detailed revision in the revised manuscript with changes marked (line 116-118, 131-133).

Point 5 and 6: The section on zircon should go in a Methodology section. missing – the remarks on zircon (p.6) don’t give enough details on the method (instrument used? Standards? QA-QC?), and where is the method used to perform whole rock and isotopic analyses described?

Response 5 and 6: Thank the reviewer for this valuable comment. The section on zircon was put into the Analytical methods section. In addition, we have also expanded details on the methods (Zircon LA-ICM-MS U–Pb dating, In-situ Zircon Lu–Hf isotopes, Major and trace elements, and Whole-rock Sr–Nd–Pb isotopes). You can find in Appendix 1. Analytical methods of the Qiaomaishan intrusions.

Point 7 and 13: Section 4.1 – ask yourself if all these diagrams are necessary / numbers can be presented in a table to avoid sentences with 10s of numbers. The statements on adakite should be discussed in the discussion section. Harker diagrams should be shown and described in the Results section.

Response 7 and 13: Thank the reviewer for these constructive comments. According to the comments of reviewers, we removed Figure 5c Zr/Ti Vs. SiO2, owing to it has same meaning with Figure 5a. In addition, the description of Harker diagrams was put in the section 4.1 Whole-Rock Geochemistry. Moreover, the statements on adakite also find in discussion section. You can find in Figure 5, line 215-221, 293-295.

Point 8: Section 4.2. – very short for a section. Define Ab and An (or use albite etc. in the text). Last sentence is an interpretation (a poorly sustained one) that has nothing to do in a Discussion section.

Response 8: Thank the reviewer for this valuable suggestion. Really, section 4.2. very short for a section. However, we only reported EMPA results of plagioclase, and this section cannot be merged into other sections. Therefore, the title of this section was revised to “4.2. Plagioclase chemistry”. In discussion 5.1 line356-358, the Ca, Fe or Mg contents of plagioclase abruptly increase from core to rim, an indication that new magma injected into magma chamber.

Point 9: Figures should be mentioned in the text before being shown / Was Ti-in-zircon calculated by reference #40? (or was it the #40 method that was used?) / move the interpretations to the Discussion section

Response 9: Thank the reviewer for these constructive comments. Ti-in-zircon temperatures were calculated by Watson et al., (2006), and this result contributes to oxygen fugacity indirectly in Figure 14b. Thus, we describe Ti-in-zircon temperatures in section 4.3. Zircon results.

Reference:

Watson, E.; Wark, D.; Thomas, J. Crystallization thermometers for zircon and rutile. Contrib. Mineral. Petrol. 2006, 151, 413-433.

Point 10 and 12: Discussion – needs to be completely re-organized. Results should be interpreted first and lead to a model, and ideas should be re-grouped (one section on sources, one of magma mingling etc.) when you say a model is not suitable, first demonstrate it and explain why you say that. delamination and any other source process have nothing to do with enclaves observed in an upper-crust magma chamber. There are source processes, deep magmatic systems (where most chemical differentiation occur) and the textures acquired late in the upper crust. Please get back to igneous petrology basis before re-writing this section.

Response 10 and 12: Thank the reviewer for these valuable suggestions. In the discussion, results of the Qiaomaishan were interpreted first (Line 286-292), and the granodiorite porphyry were regarded as adakitic rocks. The major models were discussed to be suitable for explaining the formation of Qiaomaishan adakitic rocks. These model were rule out due to multiple factors, including petrology basis, geochemical and isotopic features, or geodynamical model. As discusses above, our study tend to magma mixing model as the most feasible interpretation for the genesis of the ore-bearing granodiorite porphyry, and main evidence were shown in line 348-371. In section 5.2, we discussed in detail source characteristics of the two end-members. We proposed that granodiorite porphyry in Qiaomaishan was the product of the mixing of a felsic magma derived from partial melting of lower continental crust and a mafic magma generated from metasomatized enriched lithosphere mantle.

In delamination model, the melts derived from subsequently delaminated LCC would have acquired high Mg# signature by the interaction with the peridotite mantle at upper mantle depth. However, the existence of the widespread mafic enclaves and textural disequilibrium observed in Qiaomaishan are attributed to the injections of fresh magma into a magma chamber at crustal depths.There is no symmetry in such condition.

Point 11: a short review of what Adakites are and of there significance in porphyry-style mineralization would be useful. Also, the discussion of whether these are adakite or not should be performed by referring to the many authors that worked on adakites

Response 11: Thank the reviewer for this constructive comments. Indeed, a short review of what Adakites are and of there significance in porphyry-style mineralization would be useful. However, in the Middle-Lower Yangtze River Metallogenic Belt (MLYRB), the origin of magma and petrogenesis for the adakites related with the first episode of Cu polymetallic mineralization, based on features of geochemistry, isotopes, geodynamic background. This will repeat some models of Adakite reviews. And the discussion of whether these are adakite on account of the major models have been proposed in the MLYRB.

Point 14: (1) please use typical vocabulary – mingling, magma chamber replenishment, … / (2) make sentences / (3) English issue – this section is impossible to understand / (4) not sure I get that one – did zircon formed in the mafic magmas? Or have zircons been exchanged between the felsic and mafic magmas? (what does petro and chemistry suggest?)

Response 14: Considering the reviewer’s suggestion, we tried our best to improve the English writing and contents, made extensive changes in the manuscript. We prefer zircons were formed in mixed magma chambers. Two types of intrusive rocks have same U-Pb ages, whereas they show slightly variations in their εHf(t) values (–6.7 to –12.2, and –5.7 to –10.6, respectively), coupled with the large variations in Nd isotopic and whole-rock compositions, strong indications of magma mixing between felsic and mafic magmas during magma evolution processes. Maybe some zircons have been exchanged between the felsic and mafic magmas recorded by Hf isotopes.

Point 15: -Section 5.3.1 – to be re-worked, a lot of things don’t make sense. Cu is incompatible, it will enrich in felsic magma? May be, but there is a lot of Cu in mafic magmas too (replenishing a magma chamber with mafic melt may add a lot of Cu to the system). W should be discussed using new data (how much W is there in the studied rocks?)

Response 15: Thank the reviewer for these constructive advices. In section 5.3.1, we discuss about Cu’s behavior under high oxygen fugacity, and we suggest that lot of Cu enriched in mafic magmas derived from enriched lithospheric mantle under high oxygen fugacity (Mungall, 2002; Sun et al., 2010, 2011, 2017; Lee et al., 2012, 2020). The concentration of W in the mantle is 0.13 ppm, much lower than in the crust (upper crust is 2.0 ppm and it is 0.7 ppm in the lower crust; Arevalo et al., 2008). However, due to limitations of analytical methods for trace elements, we did not analyze the W content in our studies.

Reference:

Arevalo Jr, R.; McDonough, W.F. Tungsten geochemistry and implications for understanding the Earth's interior. Earth Planet. Sci. Lett. 2008, 272, 656-665.

Mungall, J.E. Roasting the mantle: Slab melting and the genesis of major Au and Au-rich Cu deposits. Geology 2002, 30, 915-918.

Lee, C.-T.A.; Luffi, P.; Chin, E.J.; Bouchet, R.; Dasgupta, R.; Morton, D.M.; Le Roux, V.; Yin, Q.-z.; Jin, D. Copper systematics in arc magmas and implications for crust-mantle differentiation. Science 2012, 336, 64-68.

Lee, C.-T.A.; Tang, M. How to make porphyry copper deposits. Earth Planet. Sci. Lett. 2020, 529, 1-11.

Sun, W.; Ling, M.; Yang, X.; Fan, W.; Ding, X.; Liang, H. Ridge subduction and porphyry copper-gold mineralization: An overview. Sci. China Earth Sci. 2010, 53, 475-484.

Sun, W.; Zhang, H.; Ling, M.-X.; Ding, X.; Chung, S.-L.; Zhou, J.; Yang, X.-Y.; Fan, W. The genetic association of adakites and Cu–Au ore deposits. Int. Geol. Rev. 2011, 53, 691-703.

Sun, W.; Wang, J.-t.; Zhang, L.-p.; Zhang, C.-c.; Li, H.; Ling, M.-x.; Ding, X.; Li, C.-y.; Liang, H.-y. The formation of porphyry copper deposits. Acta Geochim 2017, 36, 9-15.

Point 16: diversity??? – it is viscosity contrast that prevents mixing. What do you mean by ‘inert’? Vocabulary to be corrected – you have data on metal sources (not on mineralizing processes)

Conclusion – need to write a conclusion that looks like a conclusion

Response 16: Thank the reviewer for this valuable suggestion. Indeed, the presence of mafic enclaves and disequilibrium texture is due to viscosity contrast of magma. The mixed magma could not achieve complete homogenization. “Inert” means by W is relatively inactive in high-oxygen-fugacity hydrothermal fluid (Wood et al., 1989). Source of Cu−W polymetallic materials have been discussed in Section 5.3.1. On the basis of the magma mixing process and source of materials, we proposed most possible ore-forming process to reveal relationship between magma mixing and Cu−W mineralization in this district.

In the conclusion, we re-conclude that 2) Cu was derived from mantle magma source, whereas W was originated from LCC-related source region in this district. Magma mixing between mafic and felsic sources is the main mechanism for the formation of the particular association with Cu−W mineralization. You can find in Line 465-467.

Reference:

Wood, S.A.; Vlassopoulos, D. Experimental determination of the hydrothermal solubility and speciation of tungsten at 500° C and 1 kbar1, 2. Geochim. Cosmochim. Acta 1989, 53, 303-312.

In addition, we checked numerous typos and poor formatting in the Manuscript, Figs, and Appendix Tables. And here we did not list the changes but you can find the detailed revision in the revised manuscript with changes marked.

We appreciate for review warm work earnestly, and hope that the correction will meet with approval.

Once again, thank you very much for your comments and suggestions.

Round 2

Reviewer 2 Report

I won't be able to provide a review for this paper as long as the English has not been improved.

Author Response

Response to Reviewer 2 Comments

Dear reviewer:

Thank you for this constructive comment. Those comments are all valuable and very helpful for revising and improving our paper, as well as the important guiding significance to our research. We have studied comments carefully and have made correction which we hope meet with approval. Revised portion are marked in red in the paper. The main corrections in the paper and the responds to the comments are as following:

Point 1: Title – something is missing from the first part (The role of magma mixing… reads better) – also, please consider shortening this title

Abstract – define acronyms (MLYRB etc.), what do you mean by ‘middle-scale’? Also, abstracts need to be shortened, it needs to be organized as an abstract (with introduction, problem, objectives, methodology, etc.) and it needs to be written in English (the language used here is really hard to understand – please employ a native English speaker to proof this manuscript).

Response 1: Thank the reviewer for these constructive comments. We simplified the title and abstract to make it more consistent with the research. Title-“The important role of magma mixing in generating the Cretaceous granodioritic intrusions related to the Cu–W mineralization in the Xuancheng ore district, eastern China”. In abstract, MLYRB is the acronyms for “Middle-Lower Yangtze River Belt”, and “middle-scale” was revised as “middle-sized”. You can find the detailed revision in the revised manuscript with changes marked (line 2, 19-23).

Point 2: 2 – Cu-S skarn – is S a commodity?

Introduction – needs to be re-worked so the problem and objectives can be better explained; e.g., what does mafic enclaves have to do with all this? Etc. Also, figures are best kept for the Geological Setting section.

Response 2: Thank the reviewer for this valuable comment. Li et al., (2015) described the main ore types are copper ores, sulfur ores, copper-sulfur ores, and iron ores; and pyrite, chalcopyrite, bornite, and magnetite are main ore minerals. They have been regarded as a typical Cu–S skarn deposit before the scheelite was identified in sulfide ores. The abundance of mafic microgranular enclaves (MMEs) in the ore-formation related granodiorite porphyry. We carried out systematic geochemical and comparative studies of two types of ore-bearing intrusive rocks in Qiaomaishan, pure granodiorite porphyry and granodiorite porphyry with MMEs, to reveal the petrogenesis and possible enrichment mechanisms behind Qiaomaishan Cu–W mineralization (line 63-70). Geological Setting section is simplified, and the digital informations are omitted, due to confidentiality requirements for No.322 Unit of Bureau of Geology and Mineral Exploration of Anhui Province.

Reference:

Liu, X.; Duan, L. Geological features and Metallogenic regularity of the Tongshan-Qiaomaishan Cu-S-W-Fe polymetallic ore deposit in Xuancheng City. Geol. Anhui. 2015, 25, 174-178. (In Chinese)

Point 3: Figures 3-4 – to be improved (police size and colors, arrows…)

Response 3: We changed the abbreviation of two types intrusive rocks, i.e., Granodiorite porphyry (XC01-GP) and Granodiorite porphyry with MMEs (XC08-GPM) in Qiaomaishan, due to XC01and XC08 are easily confused with sample numbers. Meanwhile, we made some changes on legend and order of the Figures, listed in Fig.3, Fig.4, Fig.5, Fig.7, Fig.8, Fig.9, Fig.10, Fig.11, and Fig.12.

Point 4: Section 3 – how many samples? Location?... / refrain using ‘Sample…:’ (write sentences instead). Don’t describe only the samples, but talk about the context (if a granodiorite was sampled, say why – what is the aerial surface occupied by the granodiorite? What is it’s cutting relationship with the mineralization etc.? – what is the context?). It is impossible to know how representative the samples are.

Response 4: Thank the reviewer for these constructive comments. The granodiorite porphyry is the main intrusion in this area. This porphyry occurs in the form of stocks or dikes and emplaces into Carboniferous limestone, which is closely related to the skarn Cu–W mineralization in the contact zone (Figure 2b). We collected a series of samples from the drill cores, including 7 granodiorite porphyries (ZK22+02, –316 to –450 m) and 5 granodiorite porphyries with MMEs (ZK18+02, –107 to –337m), from the contact zone between Carboniferous limestone to the granodiorite porphyry. You can find the detailed revision in the revised manuscript with changes marked (line 116-118, 131-134).

Point 5 and 6: The section on zircon should go in a Methodology section. missing – the remarks on zircon (p.6) don’t give enough details on the method (instrument used? Standards? QA-QC?), and where is the method used to perform whole rock and isotopic analyses described?

Response 5 and 6: Thank the reviewer for this valuable comment. The section on zircon was put into the Analytical methods section. In addition, we have also expanded details on the methods (Zircon LA-ICM-MS U–Pb dating, In-situ Zircon Lu–Hf isotopes, Major and trace elements, and Whole-rock Sr–Nd–Pb isotopes). You can find in Appendix 1. Analytical methods of the Qiaomaishan intrusions.

Point 7 and 13: Section 4.1 – ask yourself if all these diagrams are necessary / numbers can be presented in a table to avoid sentences with 10s of numbers. The statements on adakite should be discussed in the discussion section. Harker diagrams should be shown and described in the Results section.

Response 7 and 13: Thank the reviewer for these constructive comments. According to the comments of reviewers, we removed Figure 5c Zr/Ti Vs. SiO2, owing to it has same meaning with Figure 5a. In addition, the description of Harker diagrams was put in the section 4.1 Whole-Rock Geochemistry. Moreover, the statements on adakite also find in discussion section. You can find in Figure 5, line 218-225, 297-300.

Point 8: Section 4.2. – very short for a section. Define Ab and An (or use albite etc. in the text). Last sentence is an interpretation (a poorly sustained one) that has nothing to do in a Discussion section.

Response 8: Thank the reviewer for this valuable suggestion. Really, section 4.2. very short for a section. However, we only reported EMPA results of the plagioclase, and this section cannot be merged into other sections. Therefore, the title of this section was revised to “4.2. Plagioclase chemistry”. In discussion 5.1 line356-358, the Ca, Fe or Mg contents of plagioclase abruptly increase from the core to rim, an indication that new magma was injected into magma chamber.

Point 9: Figures should be mentioned in the text before being shown / Was Ti-in-zircon calculated by reference #40? (or was it the #40 method that was used?) / move the interpretations to the Discussion section

Response 9: Thank the reviewer for these constructive comments. Ti-in-zircon temperatures were calculated by Watson et al., (2006), and this result contributes to oxygen fugacity indirectly in Figure 14b. Thus, we describe Ti-in-zircon temperatures in section 4.3. Zircon results.

Reference:

Watson, E.; Wark, D.; Thomas, J. Crystallization thermometers for zircon and rutile. Contrib. Mineral. Petrol. 2006, 151, 413-433.

Point 10 and 12: Discussion – needs to be completely re-organized. Results should be interpreted first and lead to a model, and ideas should be re-grouped (one section on sources, one of magma mingling etc.) when you say a model is not suitable, first demonstrate it and explain why you say that. delamination and any other source process have nothing to do with enclaves observed in an upper-crust magma chamber. There are source processes, deep magmatic systems (where most chemical differentiation occur) and the textures acquired late in the upper crust. Please get back to igneous petrology basis before re-writing this section.

Response 10 and 12: Thank the reviewer for these valuable suggestions. In the discussion, results of the Qiaomaishan were interpreted first (Line 291-300), and the granodiorite porphyry were regarded as adakitic rocks. The major models were discussed to be suitable for explaining the formation of Qiaomaishan adakitic rocks. These model were rule out due to multiple factors, including petrology basis, geochemical and isotopic features, or geodynamical model. As discusses above, our study tend to magma mixing model as the most feasible interpretation for the genesis of the ore-bearing granodiorite porphyry, and main evidence were shown in line 354-374. In section 5.2, we discussed in detail source characteristics of the two end-members. We proposed that granodiorite porphyry in Qiaomaishan was the product of the mixing of a felsic magma derived from the partial melting of the lower continental crust and a mafic magma generated from the metasomatized enriched lithosphere mantle.

In delamination model, the melts derived from subsequently delaminated LCC would have acquired high Mg# signature by the interaction with the peridotite mantle at upper mantle depth. However, the existence of the widespread mafic enclaves and textural disequilibrium observed in Qiaomaishan are attributed to the injections of fresh magma into a magma chamber at crustal depths. There is no symmetry in such condition.

Point 11: a short review of what Adakites are and of there significance in porphyry-style mineralization would be useful. Also, the discussion of whether these are adakite or not should be performed by referring to the many authors that worked on adakites

Response 11: Thank the reviewer for this constructive comments. Indeed, a short review of what adakites are and of there significance in porphyry-style mineralization would be useful. However, in the Middle-Lower Yangtze River Belt (MLYRB), the origin of magma and petrogenesis for the adakites related with the first episode of Cu polymetallic mineralization, based on features of geochemistry, isotopes, geodynamic background. This will repeat some models of adakite reviews. And the discussion of whether these are adakite on account of the major models have been proposed in the MLYRB.

Point 14: (1) please use typical vocabulary – mingling, magma chamber replenishment, … / (2) make sentences / (3) English issue – this section is impossible to understand / (4) not sure I get that one – did zircon formed in the mafic magmas? Or have zircons been exchanged between the felsic and mafic magmas? (what does petro and chemistry suggest?)

Response 14: Considering the reviewer’s suggestion, we tried our best to improve the English writing and contents, made extensive changes in the manuscript though MDPI English editing. We prefer zircons were formed in mixed magma chambers. Two types of intrusive rocks have same U-Pb ages, whereas they show slightly variations in their εHf(t) values (–6.7 to –12.2, and –5.7 to –10.6, respectively), coupled with the large variations in Nd isotopic and whole-rock compositions, strong indications of magma mixing between felsic and mafic magmas during magma evolution processes. Maybe some zircons have been exchanged between the felsic and mafic magmas recorded by Hf isotopes.

Point 15: -Section 5.3.1 – to be re-worked, a lot of things don’t make sense. Cu is incompatible, it will enrich in felsic magma? May be, but there is a lot of Cu in mafic magmas too (replenishing a magma chamber with mafic melt may add a lot of Cu to the system). W should be discussed using new data (how much W is there in the studied rocks?)

Response 15: Thank the reviewer for these constructive advices. In section 5.3.1, we discuss about Cu’s behavior under high oxygen fugacity, and we suggest that lot of Cu enriched in mafic magmas derived from enriched lithospheric mantle under high oxygen fugacity (Mungall, 2002; Sun et al., 2010, 2011, 2017; Lee et al., 2012, 2020). The concentration of W in the mantle is 0.13 ppm, much lower than in the crust (the upper crust is 2.0 ppm and it is 0.7 ppm in the lower crust; Arevalo et al., 2008). However, due to limitations of analytical methods for trace elements, we did not analyze the W content in our studies.

Reference:

Arevalo Jr, R.; McDonough, W.F. Tungsten geochemistry and implications for understanding the Earth's interior. Earth Planet. Sci. Lett. 2008, 272, 656-665.

Mungall, J.E. Roasting the mantle: Slab melting and the genesis of major Au and Au-rich Cu deposits. Geology 2002, 30, 915-918.

Lee, C.-T.A.; Luffi, P.; Chin, E.J.; Bouchet, R.; Dasgupta, R.; Morton, D.M.; Le Roux, V.; Yin, Q.-z.; Jin, D. Copper systematics in arc magmas and implications for crust-mantle differentiation. Science 2012, 336, 64-68.

Lee, C.-T.A.; Tang, M. How to make porphyry copper deposits. Earth Planet. Sci. Lett. 2020, 529, 1-11.

Sun, W.; Ling, M.; Yang, X.; Fan, W.; Ding, X.; Liang, H. Ridge subduction and porphyry copper-gold mineralization: An overview. Sci. China Earth Sci. 2010, 53, 475-484.

Sun, W.; Zhang, H.; Ling, M.-X.; Ding, X.; Chung, S.-L.; Zhou, J.; Yang, X.-Y.; Fan, W. The genetic association of adakites and Cu–Au ore deposits. Int. Geol. Rev. 2011, 53, 691-703.

Sun, W.; Wang, J.-t.; Zhang, L.-p.; Zhang, C.-c.; Li, H.; Ling, M.-x.; Ding, X.; Li, C.-y.; Liang, H.-y. The formation of porphyry copper deposits. Acta Geochim 2017, 36, 9-15.

Point 16: diversity??? – it is viscosity contrast that prevents mixing. What do you mean by ‘inert’? Vocabulary to be corrected – you have data on metal sources (not on mineralizing processes)

Conclusion – need to write a conclusion that looks like a conclusion

Response 16: Thank the reviewer for this valuable suggestion. Indeed, the presence of mafic enclaves and disequilibrium texture is due to viscosity contrast of magma. The mixed magma could not achieve complete homogenization. “Inert” means by W is relatively inactive in high-oxygen-fugacity hydrothermal fluid (Wood et al., 1989). Source of Cu−W polymetallic materials have been discussed in Section 5.3.1. On the basis of the magma mixing process and source of materials, we proposed most possible ore-forming process to reveal relationship between magma mixing and Cu−W mineralization in this district.

In the conclusion, we re-conclude that 2) Cu was derived from the mantle magma source, whereas W originated from the LCC-related source region in this district. Magma mixing between mafic and felsic sources is the main mechanism for the formation of the particular association with Cu–W mineralization. You can find in Line 475-477.

Reference:

Wood, S.A.; Vlassopoulos, D. Experimental determination of the hydrothermal solubility and speciation of tungsten at 500° C and 1 kbar1, 2. Geochim. Cosmochim. Acta 1989, 53, 303-312.

In addition, we checked numerous typos and poor formatting in the Manuscript, Figs, and Appendix Tables. And here we did not list the changes but you can find the detailed revision in the revised manuscript with changes marked.

We appreciate for review warm work earnestly, and hope that the correction will meet with approval.

Once again, thank you very much for your comments and suggestions.
